# The legacy of the extinct Neotropical megafauna on plants and biomes

Vinicius L. Dantas [1✉] & Juli G. Pausas [2]

Large mammal herbivores are important drivers of plant evolution and vegetation patterns, but the extent to which plant trait and ecosystem geography currently reflect the historical distribution of extinct megafauna is unknown. We address this question for South and Central America (Neotropical biogeographic realm) by compiling data on plant defence traits, climate, soil, and fire, as well as on the historical distribution of extinct megafauna and extant mammal herbivores. We show that historical mammal herbivory, especially by extinct megafauna, and soil fertility explain substantial variability in wood density, leaf size, spines and latex. We also identified three distinct regions (''antiherbiomes''), differing in plant defences, environmental conditions, and megafauna history. These patterns largely matched those observed in African ecosystems, where abundant megafauna still roams, and suggest that some ecoregions experienced savanna-to-forest shifts following megafauna extinctions. Here, we show that extinct megafauna left a significant imprint on current ecosystem biogeography.

[1] Institute of Geography, Federal University of Uberlandia (UFU), Av. João Naves de Avila, 2121, Uberlandia 38400-902 MG, Brazil. [2] Centro de Investigaciones sobre Desertificación, Spanish National Research Council (CIDE-CSIC), Ctra. Naquera Km. 4.5 (IVIA), Montcada, 46113 Valencia, Spain. ✉email: viniciusldantas@gmail.com

Over 10,000 years ago, a large proportion of our planet was populated by large and even gigantic mammals: the megafauna. Whereas most of these animals became extinct in the late Pleistocene and Early Holocene[1], there are important exceptions where a great diversity of large mammals still wanders–such as in Africa and Asia. These regions provide unique opportunities to understand megafauna ecology and its effects on ecosystems. Evidence suggests that consumption of plant biomass and related disturbances by African mega-herbivores can drive and maintain woodlands in alternative grassland states[2–4]. Moreover, large mammal herbivores impose limits on ecosystem susceptibility to fire (i.e., grazers) and can even influence soil fertility in the long term[4–6]. Thus, large mammal herbivores create and maintain their own grassy ecosystems[2]. Consequently, their extinction likely resulted in the replacement of many herbivory-maintained savannas by forests and woodlands, or by fire-maintained savannas, across the world[7,8].

Woody plant species living in herbivory-maintained ecosystems are characterised by morphological and physiological adaptations that reduce damage caused by large herbivores (i.e., antiherbivory defence traits)[7]. While defence traits that are disadvantageous under present conditions were likely lost in many places after megafauna extinction (as the selective pressure have changed), other traits may have persisted as anachronical features[9,10]. These anachronisms offer a valuable opportunity to understand past megafauna patterns and plant-megafauna interactions, and could provide insights on switches from open grassy ecosystems (with abundant megafauna, numerous grazers and highly defended plants) to closed canopy ecosystems (with the opposite features).

There are multiple traits by which plants defend themselves from large herbivores[5,7,11–13], and the dominant traits often differ with climate and availability of soil resources[12–18]. For instance, two broad savanna ecosystem types (hereafter called 'antiherbiomes', as analogous to biomes) have been recognised in Africa based on divergence in plant defence attributes and strategies, and on their typical association with specific environmental conditions[13,14,19]: (1) arid nutrient-rich savannas dominated by woody plants defended mostly with physical defences, such as thorns, small leaves and densely branched crowns (but also nitrogen-based chemical leaf defences); and (2) mesic nutrient-poor savannas dominated by plants displaying larger leaves that rely on leaf defences (e.g., leaf spines, acid detergent compounds, and lignin). While the causes of these trait-environment associations had not been rigorously addressed, evidence points towards climate and soil as key factors mediating herbivore activity due to their influences on plant tissue nutritional quality[20]. In forest ecosystems, in contrast, plants are usually less defended against megafauna, as the high primary productivity enables quick canopy escape[7]. However, forest trees often allocate resources to defences that are effective against insects and small mammal browsers (e.g., chemical defences), the main herbivores of these ecosystems[12,21,22], which suggests a different (i.e., a third) antiherbiome. Wood density was recently suggested to be a key antiherbivory resistance trait in savannas, by protecting the stems and branches against breakage and other damage by large herbivores[7]. However, it is less clear how it is related to aforementioned antiherbiomes.

Given the long history of megafauna presence and the short history of its absence, we hypothesised that the distribution of extinct megafauna left an imprint on current patterns of plant functional trait geography. We tested this hypothesis for a wide region that includes South and Central America, i.e., the Neotropical biogeographic realm. We predicted that the distribution of defence traits in the Neotropics would broadly coincide with the historical distribution of large mammal herbivores in this region. Moreover, we expected that we would be able to recognise antiherbiomes matching the ones previously recognized for Africa, within which plant assemblages converge in defensive strategies, most likely due to specific environment-herbivore interactions. If the patterns are similar to those observed in Africa, then mismatches between vegetation function and (current) structure in relation to Africa could be used to get insights on recent biome dynamics in the Neotropics.

In this work, we compiled data on plant defence traits and historical megafauna distribution to study the legacy of the extinct Neotropical megafauna on plant and biome geography. Specifically, we compiled data on spinescence (leaf and stem), leaf size, latex production and wood density for woody species and then scaled up these traits to the ecoregion[23] using species distribution data. Ecoregions are regions characterized by distinct natural assemblages of plant and animal species whose boundaries are defined based on detailed vegetation and biodiversity surveys, as well as expert opinion[23]. We correlated this data with pre-historical (from 130,000 to 1500 CE) extinct megafauna and extant mammal herbivore richness, mean body mass, and dominant diet type. In this study, the term megafauna refers to large mammal herbivores over 50 Kg that became extinct in pre-historical times (before 1500 CE), whereas the term extant mammal herbivore refers to both extant and recently extinct (after 1500 CE) species, regardless of body mass (see Methods for details). When relating plant defence traits to megafauna history, we also considered alternative hypotheses, i.e., direct associations with climate, soils, and other natural disturbances (fire and hurricanes). We also used these later variables to characterize antiherbiomes and compare their features with those reported for African ecosystems. Finally, we evaluated the extent to which former megafauna-rich savanna ecoregions have shifted to current forest-dominated ones in the Neotropics using information on megafauna richness and diet, antiherbiome classification, and published pollen fossil data for Pleistocene/Holocene vegetation.

Here, we show that megafauna history explains a large fraction of the current geographical variability in plant defence strategies. We also show that interactions among megafauna, woody plants and soil fertility shaped South-Central America and Africa similarly, allowing the recognition of the same antiherbiomes in the two continents. These antiherbiomes allow the recognition of regions likely to have experienced savanna-to-forest biome shifts in the Neotropics following megafauna extinctions. Overall, these results support the hypothesis that large mammal herbivores are important drivers of plant and ecosystem geography and that their effects can persist for a long time.

## Results

**Drivers of megafauna patterns.** We found diet information for a total of 53 megafauna species out of the 66 extinct Neotropical megafauna species in the PHYACINE dataset[24]. Based on this information, we classified the Neotropical megafauna species as browsers (22 species), grazers (16 species), and mixed feeders (15 species) (see Methods for details). The overall and per diet type megafauna richness were strongly positively correlated (Supplementary Table 1), and were highest in south-central South America (e.g., Gran Chaco in northern Argentina/southern Paraguay, grassland-forest mosaics in southern Brazil; Fig. 1a, c, d). The richness of megafauna species (overall and within each diet class) was higher in cation-poor soils and in ecoregions experiencing frequent fires, as well as for low and less seasonal rainfall areas, and was lower in islands (Supplementary Tables 2–5). Mean megafauna body mass (our proxy for body size per ecoregion) was highest in Central America and in the western part of South America,

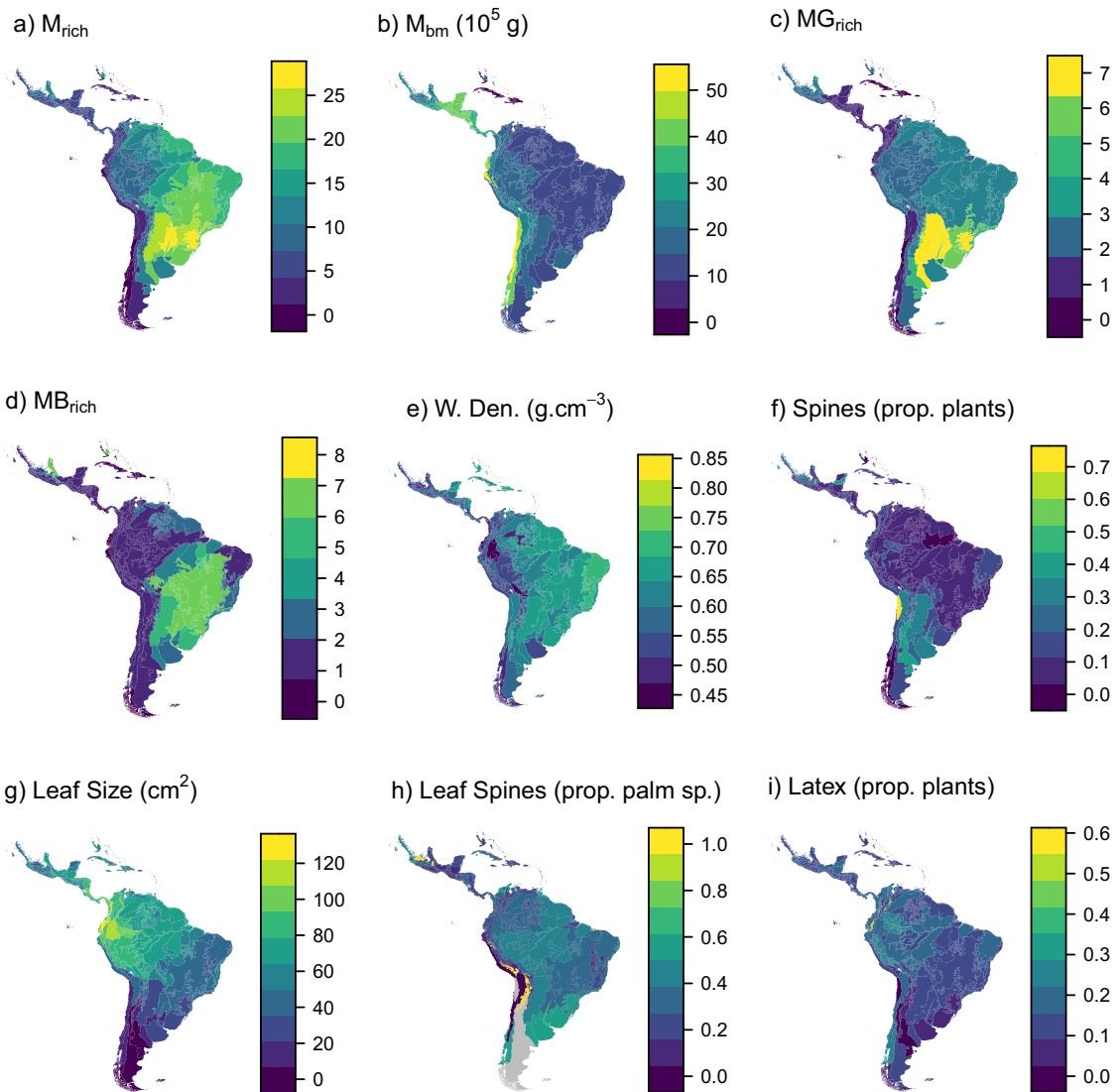

**Fig. 1 Geographical variation in extinct megafauna herbivore species distribution (a–d) and antiherbivory resistance traits (e–i) across Neotropical ecoregions.** Megafauna indicators: **a** Mean extinct megafauna species richness ($M_{rich}$); **b** extinct megafauna species mean body mass ($M_{bm}$); **c** and **d** mean extinct mega-grazer ($MG_{rich}$) and mega-browser ($MB_{rich}$) species richness, respectively (mixed feeders excluded). Antiherbivory resistance traits (**e–i**) wood density (W. Den); **e** Spines in wood tissues (**f**) leaf size (**g**) leaf spines (**h**) and latex production (**i**) Grey areas in **h** are ecoregions lacking data due to palm sensitivity to frost. prop.: proportion; sp: species.

following a longitudinal gradient (Fig. 1b), and was especially high in moist habitats, seasonal climates, and fertile soils; in contrast, it was extremely low on islands (Supplementary Table 2–5).

**Megafauna history and antiherbivory defence traits.** Most of the studied plant traits had substantial variability explained by megafauna history and some had more variability explained by megafauna (e.g., wood density, leaf spines) than by any other factor (Fig. 2; Supplementary Tables 2–4). Wood density and leaf spines increased, whereas leaf size decreased, with megafauna richness (Fig. 2a, c, d; Supplementary Table 3). Stem spines was only affected by megafauna body mass, and was promoted by mid-size megafauna (significant quadradic term; Fig. 2b; Supplementary Table 3). Extant mammals were also important for this trait, but soil pH was the most important factor, with more spinescent plants occurring in the most alkaline soils (Fig. 2b; Supplementary Table 3). The occurrence of more extinct megafauna grazers relative to browsers was an important driver of investments in defence such as wood density and spines (Fig. 2; Supplementary Tables 3 and 4),

highlighting the important role of typical savanna megafauna. The only trait that was not influenced by megafauna was latex, which was favoured by small extant mammal herbivores and by acidic soils with relatively high cation exchange capacity. For each trait, we compared the coefficients describing their relationships with the above faunal indicators and those obtained based on trait values derived from 1,000 randomizations of the plant species abundance by ecoregion matrix. In none of the cases, the original coefficient values were within the range between the 0.05–0.95 quantiles of the random coefficients (Supplementary Table 5). A small (but significant; $\Delta$AIC > −2) improve in AIC was observed when $M_{rich}$ was replaced by megafauna browser richness ($MB_{rich}$) or by megafauna density in the selected models of Wood Density and Leaf Spines, and, when replacing $M_{rich}$ by $MB_{rich}$, for Leaf Size (Supplementary Table 6).

**Antiherbiomes.** We found three principal component axes (PCA) of antiherbivory defence strategy. The first PCA axis segregated assemblages in which plants allocated resources to physical vs.

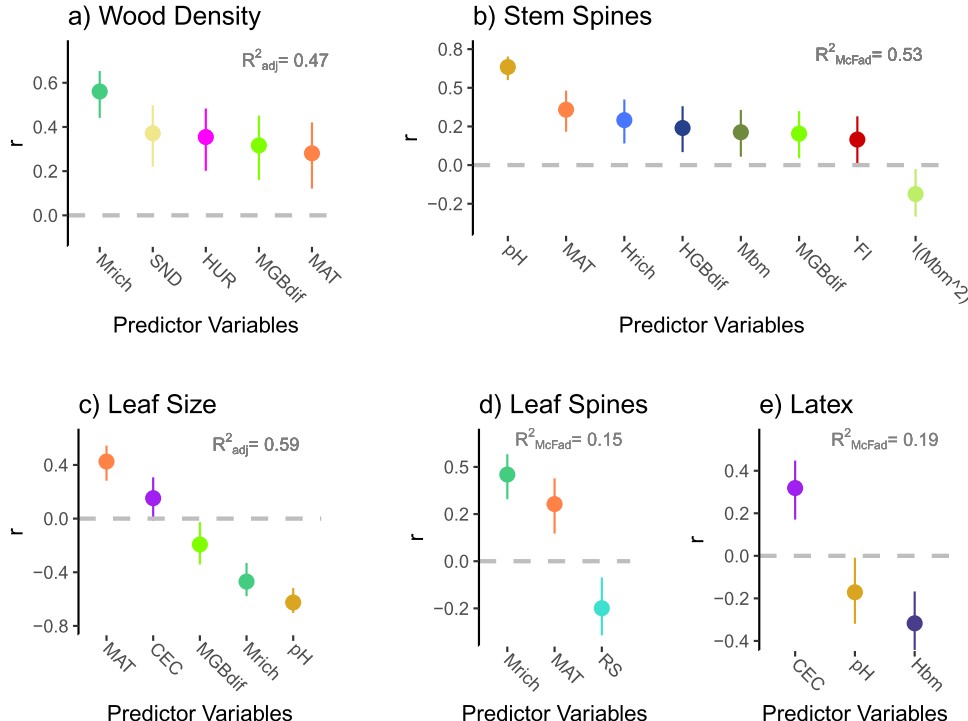

**Fig. 2 Effect sizes (Pearson's *r*; dots) and 95% confidence intervals (vertical bars) for each predictor variable in the selected regression models of each plant defence trait (a–e; see Supplementary 2–4 for detailed results).** Model's adjusted $R^2$ ($R^2_{adj}$) and McFadden's pseudo-$R^2$ ($R^2_{McFad}$) are shown for continuous and binary traits, respectively. $n = 143$ ecoregions (except for Leaf Spines, for which $n = 132$ biologically independent ecoregions). Islands and ecoregions with incomplete data were excluded from all analyses. Mrich: mean extinct megafauna species richness; Mbm: mean extinct megafauna species mean body mass (Mbm^2: Mbm squared); MGBdif: richness difference between grazer and browser megafauna (mixed feeders excluded); Hrich: extant mammal herbivore richness; HGBdif: richness difference between extant mammal grazers and browsers; Hbm: mean body mass of extant mammal herbivores; MAT: mean annual temperature; RS: rainfall seasonality; CEC: soil cation exchange capacity; pH: soil pH; SND: soil sand percentage; FI: fire intensity; HUR: hurricane counts.

chemical defences (named "Physical-Chemical"; first PCA axis; accounting for 37.8%; Fig. 3b, c). The second was related to the presence of leaf spines (named "Leaf Spine Axis"; 21.8%; Supplementary Fig. 1), whereas the third was related to stem defence mechanism: spines vs. wood density (named "Thorny-Woody"; 17.8%; Fig. 3b, c). A hierarchical clustering analysis on these axes, supported the existence of three distinct antiherbiomes in the Neotropics (Fig. 3a, c; Table 1); only the first and third axes significantly differentiated these antiherbiomes (Kruskal-Wallis $\chi^2$ of 106.58 and 81.88, respectively; both $P < 0.001$; Fig. 3d and e; for axis 2: Kruskal-Wallis chi-squared = 5.024; $P = 0.081$; Supplementary Fig. 1). The Small Leaves Thorny (SLT) antiherbiome was characterized by thorny and small-leaved species, as well as arid, cold and nutrient-rich ecosystems, containing numerous extinct and extant large grazers (Figs. 3d, e and 4; Supplementary Table 7). The Broad Chemically-defended Leaves (BCL) antiherbiome was characterized by plants with extremely large and chemically defended leaves, and was mostly associated with moist climates and intermediate fertility soils (relatively high cation exchange capacity and low pH) and few but large megafauna species, especially browsers (Figs. 3 and 4; Supplementary Table 7). Finally, the Intermediate Leaves Woody (ILW) was characterized by intermediate leaf sizes and levels of chemical defences, and very high wood density, as well as by a high megafauna richness, especially in relation to small browsers and mixed-feeders, by moist and hot climates, and extremely nutrient-poor soils (Figs. 3 and 4; Supplementary Table 7). While these differences were significant, regression analyses on the Physical-

Chemical and Thorny-Woody axes indicated that soil fertility and megafauna history were the main drivers of these patterns, with extant grazers playing a minor but significant role (Figs. 5 and S2; Supplementary Tables 2–4). Replacing megafauna richness by the richness of browsers did not improve model fit for these axes, although replacing it by megafauna density (estimated using allometric equations and body mass; Supplementary Fig. 3; see Methods for details) significantly reduced AIC for the Thorny-Woody axis (Supplementary Table 6). The BCL, ILW and SLT antiherbiomes corresponded to 26, 55 and 20% of the total Neotropical biogeographic realm area (Table 1). Most of BCL and ILW areas consist of ecoregions currently classified as Tropical & Subtropical Moist Broadleaf Forest biome, whereas most of the SLT corresponds to Temperate Grasslands, Savannas & Shrublands (Table 1).

**Biome shifts**. We identified 27 forest-dominated ecoregions with evidence of having been formerly dominated by grassy ecosystems (i.e., savanna or grassland) during the Pleistocene. These ecoregions cover an area of 6,407,594 Km$^2$ (Fig. 6), within which 4,469,901 Km$^2$ are currently moist forest (70%) and 1,937,693 Km$^2$ dry forest and woodlands (30%). These areas were identified based on their extremely high richness of megafauna and mega-grazer species (i.e., greater than the 0.75 quantile for these two groups), and because they were classified in the SLT or ILW antiherbiomes (for which megafauna and trait patterns suggested that they used to be grassy ecosystems). Using these two criteria, we also found that all of the currently savanna-

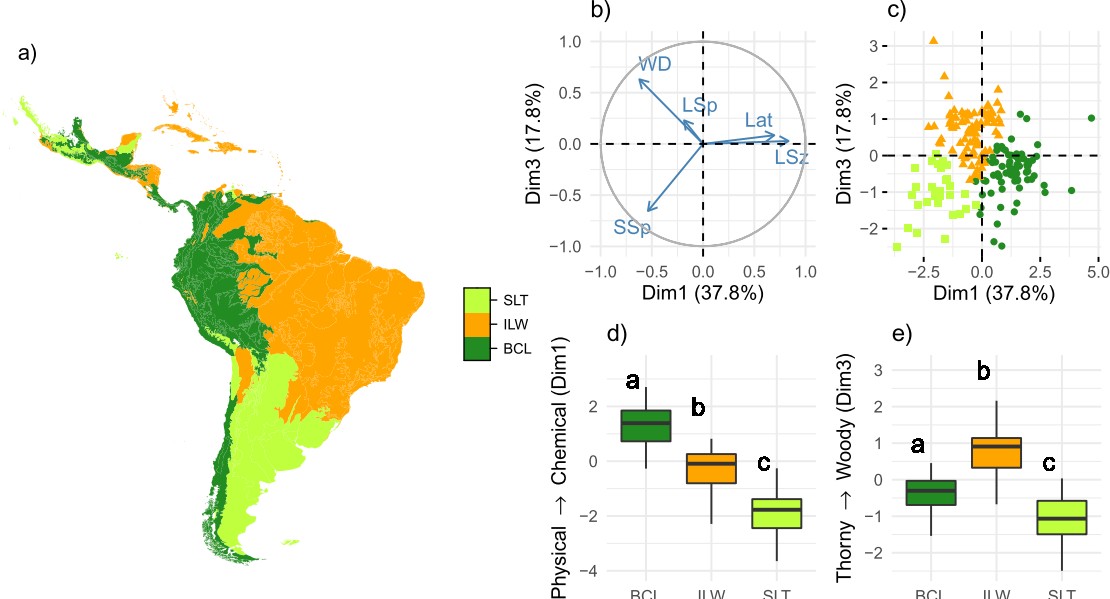

**Fig. 3 Geographical distribution (a) and functional trait characterisation (b–e) of Neotropical antiherbiomes.** Antiherbiomes were classified using hierarchical clustering on principal component axes (PCA; **b** and **c**) of mean antiherbivory defence trait values per ecoregion. Antiherbiomes are: SLT: Small Leaves Thorny (light green); ILW: Intermediate Leaves Woody (orange); BCL: Broad Chemically-defended Leaves (dark green; see Results for details). Only PCA dimensions 1 (Dim1) and 3 (Dim3) significantly differed among antiherbiomes (see Supplementary Fig. 1 for Dim2), representing a physical-to-chemical defence (Dim1; **d**) and a wood density-to-stem spine (Dim3; **e**) axis, respectively. Different letters in the boxplots indicate significant differences among antiherbiomes ($n = 150$ ecoregions; $P = 0.000$ for global and pairwise comparisons, except for BCL vs. SLT in **e** in which $P = 0.007$; $P$-values corrected using the Benjamini & Hochberg method; islands excluded; see Supplementary Table 7 for details). LSz: leaf size; SSp: stem spines; WD: wood density; LSp: leaf spines; Lat: latex. Boxplot description: center line, median; box limits, first and third quartiles; whiskers, 1.5x interquartile range; outliers not shown.

**Table 1 Antiherbiome area per biome as a proportion of (divided by) the total area of the Neotropical biogeographic realm.**

| Biome | Antiherbiome | | |
|---|---|---|---|
| | BCL | ILW | SLT |
| Deserts & Xeric Shrublands | 0.01 | 0.00 | 0.01 |
| Flooded Grasslands & Savannas | 0.00 | 0.01 | 0.00 |
| Mangroves | 0.00 | 0.01 | 0.00 |
| Mediterranean Forests, Woodlands & Scrub | 0.01 | 0.00 | 0.00 |
| Montane Grasslands & Shrublands | 0.01 | 0.01 | 0.02 |
| Temperate Broadleaf & Mixed Forests | 0.02 | 0.00 | 0.00 |
| Temperate Grasslands, Savannas & Shrublands | 0.00 | 0.00 | 0.08 |
| Tropical & Subtropical Coniferous Forests | 0.01 | 0.00 | 0.00 |
| Tropical & Subtropical Dry Broadleaf Forests | 0.01 | 0.08 | 0.05 |
| Tropical & Subtropical Grasslands, Savannas & Shrublands | 0.01 | 0.14 | 0.02 |
| Tropical & Subtropical Moist Broadleaf Forests | 0.17 | 0.29 | 0.01 |
| Total | 0.26 | 0.55 | 0.20 |

Biomes modified from Dinerstein et al.[48] (i.e, the Dry Chaco ecoregion is here included in the Tropical & Subtropical Dry Broadleaf Forests). Antiherbiomes are: SLT: Small-Leaves Thorny; ILW: Intermediate Leaves Woody; BCL: Broad Chemically-defended Leaves.

dominated ecoregions of South America were also savanna in the past (i.e., were Stable Savannas), occupying 3,725,492 km². In these later areas, we also included the Llanos ecoregion (375,787 km²; 12%), despite of having one species less than the 75% quantile of megafauna species richness (i.e., 13 rather than 14 megafauna species).

We found 22 fossil sites with evidence of past savanna dominance, 15 of which were located in currently forest-dominated ecoregions identified to have experienced savanna-to-forest shift (eight ecoregions in total; Fig. 6; Supplementary Table 9); the other seven were in currently savanna-dominated ecoregions that were identified to be stable savannas (i.e., Llanos, Cerrado and Campos Rupestres ecoregions; Fig. 6; Supplementary Table 9). Among the 15 sites in forest-dominated ecoregions, two were located in areas that currently consist of savanna patches within the Amazon forest region (i.e., in the Purus-Madeira moist and Maranhão Babaçu forest ecoregions; Supplementary Table 9), whereas the remaining 13 were found in areas that are also forest at the local scale (Fig. 6; Supplementary Table 9). From the seven sites located in savanna-dominated ecoregions, three are currently forest at the local scale, all of which are located in areas of transition with a forest-dominated ecoregion hypothesized to have experienced savanna-to-forest biome shifts. Thus, a total of 16 fossil sites (out of the 22; i.e, 72%) support savanna-to-forest shift consistent with our initial hypothesis, 13 sites within and three sites nearby (i.e., transitional areas) the identified shifting forest ecoregions.

## Discussion

Historical herbivory was among the most important predictors of antiherbivory defence traits (Fig. 2) and strategies (Fig. 5) in the Neotropics. Some plant defence traits (e.g., wood density, leaf spines) were better explained by geographical patterns associated with extinct megafauna than by extant herbivore species; although the latter were important drivers of stem spines (total and grazer extant mammal richness; Fig. 2b) and latex production (smaller extant mammals; Fig. 2e). These results supported the idea that mammal herbivores, especially large ones, explain an important proportion of the global variation in plant traits[7]. They also suggest that the consequences of the selective effects of megafauna on plant assemblages last for millennia. More

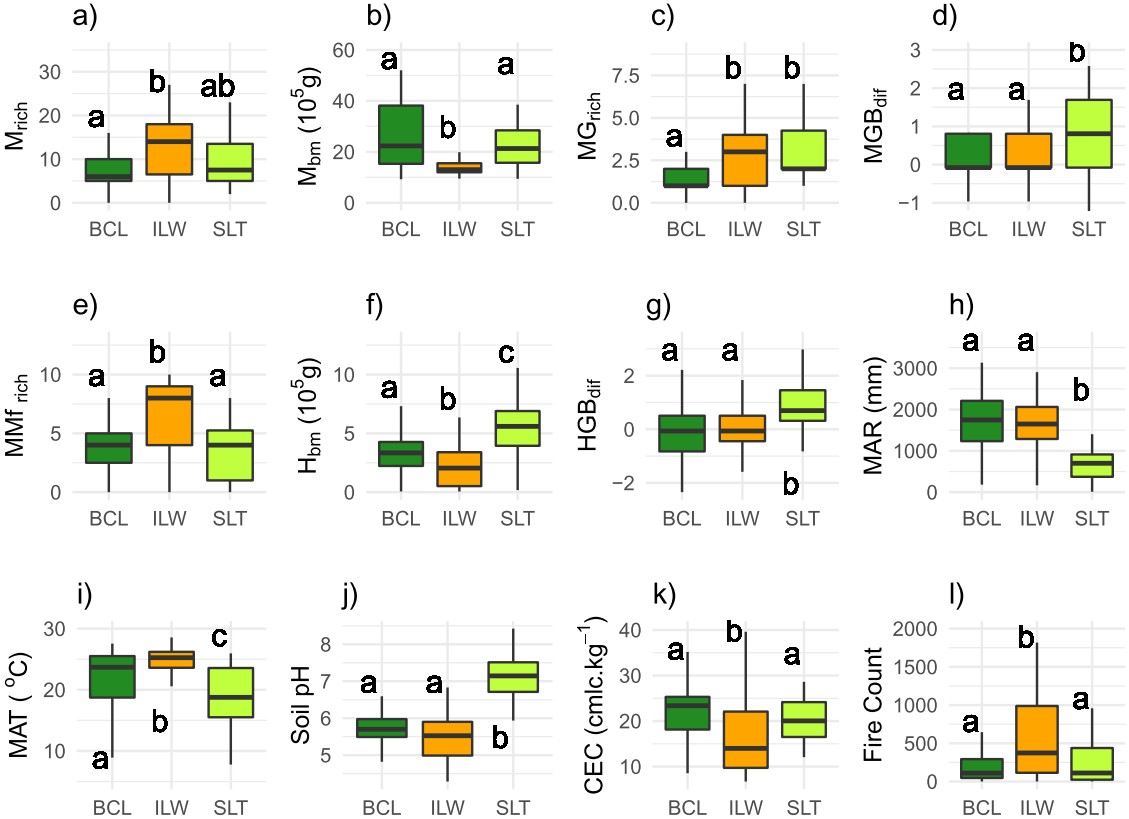

**Fig. 4 Megafauna (a–e), extant mammal (f–g), environmental (h–k), and fire regime (l) differences among the three Neotropical antiherbiomes (SLT: Small-Leaves Thorny, light green; ILW: Intermediate Leaves Woody, orange; BCL: Broad Chemically-defended Leaves, dark green).** Different letters indicate significant differences among antiherbiomes (Kruskall-Wallis test, followed by Dunn pairwise comparison; *P*-values corrected using the Benjamini & Hochberg method; islands excluded; non-significant variables omitted, see Supplementary Table 7). $M_{rich}$: extinct megafauna species richness; $M_{bm}$: extinct megafauna species mean body mass; $MG_{rich}$: mega-grazer richness; $MGB_{dif}$: difference between the number of extinct megafauna grazers and browsers; $MMf_{rich}$: mega-mixed feeder richness; $H_{bm}$: Body mass of extant herbivore species; $HGB_{dif}$: difference between the number of extant mammal grazers and browsers; MAR: mean annual rainfall; MAT: mean annual temperature; CEC: soil cation exchange capacity. The richness of megafauna browsers also differed between the ILW (higher) and the BCL (Supplementary Table 7). Boxplot description: center line, median; box limits, first and third quartiles; whiskers, 1.5x interquartile range; outliers not shown. $n = 150$ ecoregions for $M_{rich}$, $MGB_{dif}$, $MG_{rich}$, $MB_{rich}$, $MMf_{rich}$ and $HGB_{rich}$; 145 for $M_{bm}$ and $H_{bm}$; 146 for Fire Count, CEC and pH; and 148 for MAR and MAT. Global *P*-values equal zero at three decimal digits to all variables, except for $MGB_{dif}$ (0.019), $MB_{rich}$ (0.002) and Fire Count (0.001) (pairwise *P*-values and other statistics provided in Supplementary Table 7).

generally, these findings highlight the key role of history, natural disturbance regimes, and species interactions for understanding global patterns of plant functional trait variability[7].

While megafauna richness was the most important predictor of wood density and leaf spines, and of the Thorny-Woody axis, soil pH was the most important predictor of stem spines and leaf size, and of the physical vs. chemical defence axis (Figs. 2 and 5; Supplementary Table 3). Soil fertility generally increases with soil pH, and evidence from African savannas suggests that high soil fertility results in increased large mammal herbivore density[13,25]. Unfortunately, there is a lack of reliable data on extinct megafauna species density, and the main currently available methods to estimate this density are based on body mass, resulting in a strong positive correlation with megafauna richness (Supplementary Fig. 3). Cation exchange capacity (CEC) is another indicator of soil fertility. However, the high acidity and aluminium content of the clay soils that dominate in much of the study area (i.e., the tropical region) makes cations largely unavailable for plants and, thus, CEC, alone, is often a poor indicator for soil fertility. A high soil pH generally implies a high abundance of plants whose leaves are nutrient-rich and, therefore, preferred by herbivores[13,21]. The idea that soil pH effects on these traits are mediated by herbivory is consistent with previous evidence that, in Africa, the effects of

climate and soil on plant nutritional quality mediate the distinction between arid nutrient-poor savannas, with spinescent small leaved plants, and mesic savannas with the opposite features[20]. In fact, soil pH is a key feature differentiating arid-eutrophic (high pH) and mesic-dystrophic (low pHs) savannas in Africa[19]. Plants with N-fixer symbionts, which are expected to present N-rich leaves, are also especially favoured in alkaline soils (also under high temperatures)[26]. Accordingly, in African savannas, traits like abundant stem spines, small and N-rich leaves are correlated[13,14]. In addition, while legumes (which often harbour N-fixing symbionts), corresponded to 18% of the species in our stem spines database, they accounted for 32% of all spinescent species and also had significantly smaller leaves than non-legume plants (Supplementary Fig. 4). The high investments in physical plant defences under conditions that favour the production of nutrient-rich leaves suggests that chemical leaf defences are less effective in deterring herbivores from consuming plants with high nutritional quality leaves. Thus, physical defences like spines and small leaves are necessary as additional protections to efficiently reduce leaf intake rates[14].

In addition to soil pH, megafauna richness also promoted physically defended species (Figs. 2 and 5). However, megafauna richness favoured high wood density rather than thorns (Fig. 5).

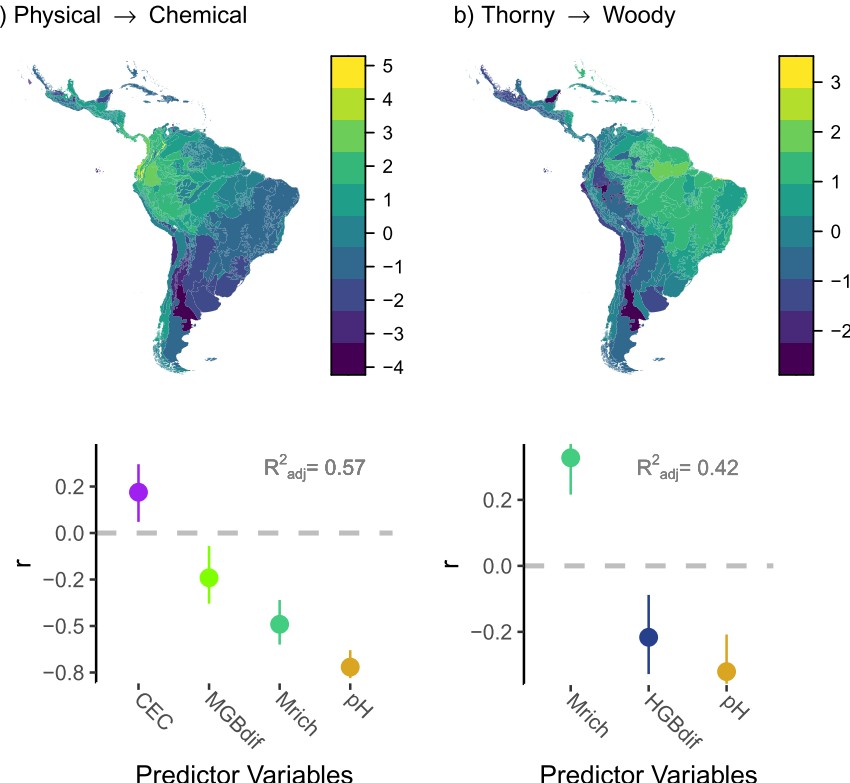

**Fig. 5 Distribution maps (upper panels) and regression results (lower panels) for the Physical-Chemical (a) and Thorny-Woody (b) strategy axes of antiherbivory defence traits (see also Fig. 3).** In the lower panels, dots are $r$ statistic values and lines represent 95% confidence intervals. $n = 150$ ecoregions for the two boxplots. Boxplot description: center line, median; box limits, first and third quartiles; whiskers, 1.5x interquartile range; outliers not shown. CEC: soil cation exchange capacity; MGB$_{dif}$: difference between the number of extinct megafauna grazers and browsers; M$_{rich}$: extinct megafauna species richness; pH: soil pH; HGB$_{dif}$: difference between the number of extant mammal grazers and browsers. Maps for the main predictors of these two axes are presented in Supplementary Fig. 3.

Stem spines tended to be favoured, instead, by an even higher soil pH, mid-size extinct megafauna species (as in African savannas[12]) and by extant mammal species (Fig. 2b). However, both wood density and stem spines were favoured in grazer-rich environments (i.e., in relation to browser richness; see Methods), suggesting that these traits are more adaptive against savanna herbivores and in grassy ecosystems than in forests (C4 grasses are generally shade intolerant[2]). In forests, a dense canopy imposes strong allocation constrains for woody plants due to the severe competition for light. Moreover, the high productivity of forests may alleviate the need for traits that protect trees from megafauna by allowing a fast canopy escape from ground dwelling animals via a fast stem elongation[2,7]. Small leaves, in contrast, seems to be especially favoured by browsing (Fig. 2c), suggesting that this trait may be also adaptive in shrublands and woodlands (although less so under hot climate; Fig. 2c).

The three detected antiherbiomes (Fig. 3) were broadly consistent with those previously reported for Africa. The SLT antiherbiome had features matching African arid nutrient-rich savannas, such as the widespread presence of small-leaved plants with thorny stems and branches, the dry climate and high nutrient availability (as indicated by both high soil pH and CEC), and was concentrated at higher latitudes (Figs. 3 and 4; [2,13,14,19,27]). Likewise, the features of the ILW resembled those of African mesic nutrient-poor savannas. Compared with the SLT, plants in these ecosystems had relatively large leaves, less spines and relied more (but less than the BCL) on chemical defences (Fig. 3; [13,14]). The climate in the ILW antiherbiome was also moister and hotter, the soils were nutrient-poorer (compared with the SLT), and fires were

very recurrent (Fig. 4). The latter also seems to be the case of mesic-dystrophic savannas in relation to arid-eutrophic ones in Africa[2,19,22,27], where higher fire activity in mesic savanna results from the dominance of tall flammable grasses instead of the short and less flammable grazing adapted grasses that dominate in arid savannas[2]. Accordingly, grazers were more common in relation to browsers in the SLT than in the ILW (Figs. 4d and g), as observed in Africa for arid (in relation to mesic) savannas[27]. However, the richness of grazers was high in both the SLT and the ILW in the Pleistocene (Fig. 4c), suggesting a savanna-like structure in both cases. Palaeontologic studies suggest that Neotropical regions associated with the SLT and the IWL also differed in megafauna species composition[28,29], consistently with the distinct megafauna species composition in these two antiherbiomes in Africa[19]. Interestingly, high wood density appeared as a key characteristic of the ILW antiherbiome (Fig. 3b and c), a pattern that have not been previously addressed in African savanna ecosystems (but see ref.[7]).

In contrast, body mass followed the opposite pattern in relation to Africa, where larger species occur in mesic nutrient-poor savannas[27] (while, here, they were associated with the SLT antiherbiome; Fig. 4). Given the lack of a body mass effect in the principal component axes separating the antiherbiomes (Fig. 5), body mass is unlikely to be an important driver of antiherbiome assembly for species larger than 50 kg. The association of large body mass with the SLT and BCL antiherbiomes could be a geographical artefact, as body mass followed a longitudinal gradient in the Neotropics (Figs. 1b and 3), but the subject deserves further investigation. Also, while the SLT antiherbiome appeared to be relatively megafauna poor (Fig. 4a), this reflected the fact

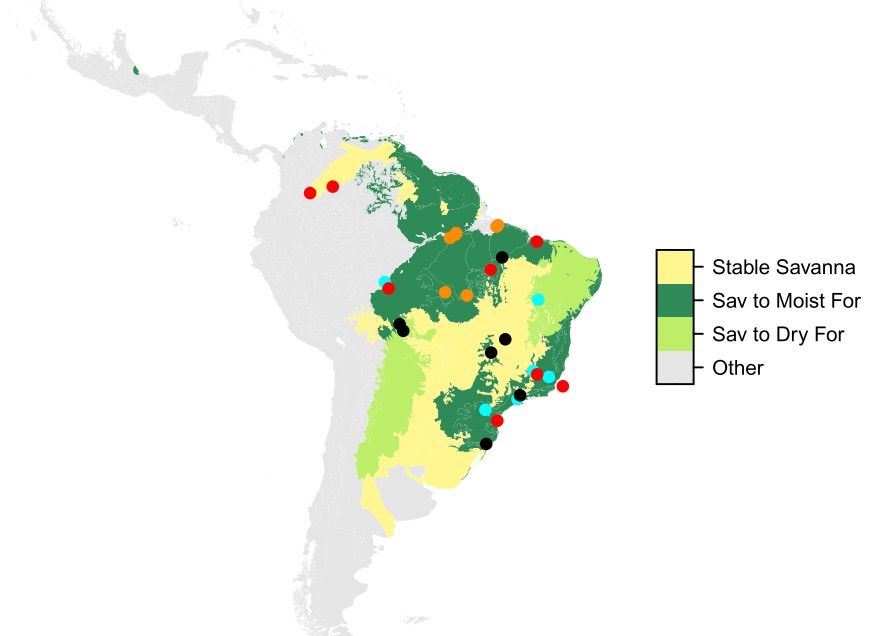

**Fig. 6 Hypothesized Pleistocene savanna region of South America (coloured areas).** These were defined as ecoregions from the SLT or ILW antiherbiomes (see Fig. 3) that were rich (richness ≥ 75% quantile) in both megafauna species and large grazers (indicating $C_4$ grass presence) in the period (Llanos savannas, with one megafauna species less, was also included: north-western most yellow patch). Yellow patches (Stable Savanna) have remained as savanna (were and still are savannas), whereas, the green areas shifted from a savanna to either a dry (Dry For; light green) or a moist (Moist For; dark green) forest/woodland state. From 22 fossil sites (non-orange dots) presenting evidence of past savanna-dominated states (red for Last Glacial Maximum; cyan for Mid-Holocene; and black for both periods), 16 (73%) are currently located in forest ecosystems at the local scale. Among these, 13 (59%) are located within and three (13%) at the vicinity of forest-dominated ecoregion with functional trait and megafauna evidence suggestive of a savanna-to-forest shift. The other six fossil sites (27%) are currently savannas at the local scale. They are located in either currently forest- (two sites or 9%; the western-most cyan and the north-easternmost red dots) or savanna- (4 sites or 18%; two easternmost cyan, the northmost red and a black dots in the large central yellow patch) dominated ecoregions (see Supplementary Table 9 for more details). Areas currently containing small (relict) savanna patches in the Amazon forest (orange dots) are also shown.

that large areas of the SLT are located in the Andes Mountains, in which the low temperatures may limit the success of both megafauna and woody plant species. Accordingly, most lowland ecoregions of the SLT were, in contrast, megafauna rich (except for the Humid Pampas; Fig. 1a). Another key difference in relation to Africa is the very broad rainfall ranges of each antiherbiome, which for the BCL and ILW comprised from arid to moist climates (Table 1), a pattern consistent with substantial climate change since the Pleistocene. If so, antiherbiome persistence over time is likely to be little influenced by climate change, and the same could apply to antiherbiome assembly. Consistent with this idea, regression analyses for the Physical-Chemical and Thorny-Woody axes suggested that only nutrient availability and megafauna history explain these large-scale patterns (Fig. 5). However, it is important to bear in mind that, in our dataset, soil pH and mean annual precipitation were negatively correlated ($r = -0.78$; Supplementary Table 1) and, thus, these two variables were not entered in the same model to avoid multicollinearity (i.e., model selection was carried separately for each, and the model with the lowest AIC was selected as the best; see Methods).

The BCL had features consistent with a long-term forest in which the high productivity of the woody layer has historically protected tree canopies against megafauna, although not from smaller mammal browsers (which includes tree climbers) and insects. Therefore, BCL plants often lack physical defences, but instead, use chemical defences (Fig. 3) which are more effective against these smaller herbivores[21] (Fig. 2e). The lack of grazers and mixed-feeder species in the BCL region during the

Pleistocene (Figs. 1c and 4), as well as the fact that the vast majority of these ecoregions are currently moist forests (Table 1), suggests long-term forest stability. This stability could be related to the high climatic stability of these areas (associated with orographic Andes-related rainfall and rainfall produced by plants within the Amazon basin), as well as the nutrient-rich sediments (high CEC) derived from the Andes Mountains (responsible for the "white" water nutrient-rich rivers of the Amazon, as opposed to nutrient-poor "black" water rivers).

Ecoregions classified within the SLT and the ILW antiherbiome currently include both forest- and savanna- dominated ones (Table 1), suggesting savanna-to-forest shifts during the Holocene. This should be especially true for forest ecoregions that used to be megafauna and mega-grazer rich during the Pleistocene (Fig. 6). From the 22 fossil sites used to validate this rationale, there was evidence of savanna-to-forest shifts for 16 (72%), as they: (1) were located within (13 sites; 59%) or nearby (i.e., in transitional zones; three sites; 13%) forest-dominated ecoregions here hypothesized to have been savannas in the past (see criteria above); and (2) occurred in areas that are currently a forest ecosystem at the local scale (Fig. 6; Supplementary Table 9). The remaining six fossil sites were and still are grassy ecosystems at the local scale, out of which two are currently located in savanna patches within a forest-dominated ecoregion (Fig. 6; Supplementary Table 9). Therefore, the available fossil data was interpreted to provide general support to our initial hypothesis that antiherbiomes distribution and megafauna patterns allow the correct identification of biome shifts in the continent.

Paleoclimate alone is unlikely to explain the replacement of most of these savannas by forests[30], suggesting, instead, that large herbivores played a key role in maintaining grassy biomes during the Pleistocene. Thus, megafauna extinctions could largely explain the current prevalence of forests in the continent. While we did not find fossil evidence supporting savanna-to-forest shifts for the Dry Chaco and Caatinga dry forests and for interior South-eastern Amazon areas, these regions (as well as those associated to Brazilian Atlantic forest) are known to contain numerous isolated savanna patches (Fig. 6), often interpreted as relicts of large savannas in the past[31–34]. In fact, whereas the Chaco and Caatinga regions are often defined as dry forest[34], their vegetations often consist of shrublands and woodlands (collectively called "woodlands" in Fig. 6) that mainly differ from savanna due to the lack of a continuous grass cover. In addition, recent studies suggest that South-eastern Amazon areas are especially sensitive to small reductions of canopy cover as this could feedback to decrease regional rainfall, partially produced by the trees themselves[35].

Some traits that are usually assumed to be driven by the effects of climate and soil[7,36,37] are here interpreted as resulting from historical herbivory. This is the case of woody density, which was mainly predicted by megafauna richness (Fig. 2a; Supplementary Table 3), and leaf size, which had megafauna richness as its second most important driver (Fig. 2c). Global studies typically evaluate the effects of climate and soil variables without considering alternative processes (e.g., natural disturbances), although, often, these variables explain only a small proportion of the trait variability[7,36,37]. Environmental variables can largely co-vary spatially with disturbance-related variables, which makes inferences based on simple correlations with a small set of variable unreliable. For instance, in a recent study in the Amazon basin[38], wood density was found to be influenced by soil variables that are shown here to correlate to megafauna patterns, such as cation availability and soil physical properties (Supplementary Table 3). Moreover, while Wright et al.[37] found mean annual rainfall (MAR) to explain leaf size, we show that MAR is strongly negatively correlated to soil pH ($R = -0.78$), which is an important driver of herbivory. While in the Amazon study above[38] correlations were often found to be stronger than in our results, this is probably because these results were based on stand-level data and considered only one biome (i.e., they were less subject to error). In any case, caution is needed when comparing studies addressing different spatial scales. Moreover, the above-mentioned leaf size study[37] also included herbaceous species, which were not considered here (see Methods for details). Despite of that, our results provide compelling evidence that the role of wood density and leaf size as antiherbivory defence traits are likely to have been under-rated in the functional ecology literature[7]. We also provide novel evidence that a high wood density is under positive selection by hurricane activity (Fig. 2a), which suggests that wood density is a key disturbance-related trait.

The reported biome switches following megafauna extinctions suggest that shifts in the selective ecological pressures experienced by plants have occurred from the Pleistocene to the present. Whereas, in mesic seasonally dry ecosystems, megafauna extinction must have increased fire activity, in browser-dominated and/or productive ecosystems, it led to reduced light availability (woody encroachment)[39]. Functional trait trade-offs probably limited the accumulation of adaptations to both ancient and novel conditions and this process is likely to explain some of the residual variability in the regression models. Moreover, the introduction of exotic megafauna species in many places (live-stock) can replace the ecosystem functions of extinct species[40], which could also shift the geography of defence traits away from

past patterns. This should be especially true for leaf traits, which are generally more plastic than stem traits[18,41]. For instance, the expression of leaf spines, the trait with the lowest percentage of variability explained by the selected models, is substantially affected by cattle density[18]. Thus, trait-megafauna associations were probably much stronger in the past. Our difficulty in tracking herbivore effects on leaf traits is probably also explained by the fact that different leaf defences often co-evolve in interaction with specific herbivore species, rather than evolving in response to generalist herbivores[21]. Moreover, some types of defences are more likely to be present in certain taxa, as seems to be the case of leaf spines, which tend to be mostly restricted to palms in the region (see Methods for details). Stem spines can also be relatively plastic in some species[14], which could further add uncertainties to our results. However, the variability in stem spines that was explained by herbivory in our results (sum of the average contributions = 13%; Supplementary Table 3) was comparable to values previously reported for African savannas (8.1%[12]; although statistical methods are different). While, in these later savannas, social mixed-feeder species are important predictors of stem spines[12], replacing the richness of megafauna species by the richness of mixed feeders in the final model did not improve the fit for this trait (i.e., AIC increased 3.56 points).

The limited availability of trait and distribution data for Neotropical megafauna and plant species could also be potential sources for low predictability in our models. Yet, plant species richness in our dataset was strongly positively correlated with previously estimated vascular plant species richness for these ecoregions[42], even for leaf spines, which only included palm species ($r = 0.65 – 0.75$). Moreover, our data often represented over 20% of the woody vascular flora of these ecoregions (see Methods for details). Thus, our dataset seems to be fairly representative of the overall underlying regional patterns. Another potential source of uncertainty in our results relates to the explanatory variables (e.g., our herbivory indicators, as well as the climate and soil variables) most of which are estimations, rather than measurements, drawing on a range of assumptions. Future studies providing more detailed data on megafauna densities and diets (e.g., allowing a more detailed differentiation of distinct guild, such as folivores, frugivorous) and on fire and environmental data could especially advance our understanding of megafauna effects. Finally, we suggest that data on herbaceous species traits could shed additional light on Neotropical anti-herbiomes by better evidencing the biogeographic influence of grazers.

Overall, we provide support to the idea that historical herbivory regimes explain a large fraction of the current biogeographic variability in plant functional traits and ecosystem geography. This effect has been largely neglected in the ecological literature[7,43]. Furthermore, we found that the interplay among herbivory regimes, climate, soil, and plant traits in large geographical and geological scales results in the emergence of globally distributed antiherbiomes characterised by plant assemblages showing convergent antiherbivory defence strategies. Once assembled, these antiherbiomes persist for millennia, despite the cascading effects of megafauna extinction on the functioning of ecosystems. These antiherbiomes represent one of the most striking broad-scale biological anachronisms and allow the detection of grassy- to forest- dominated ecosystem shifts in South America after megafauna extinction. These results highlight the importance of past and present megafauna distributions to understand plant and ecosystem functional biogeography.

## Methods

**Plant defence traits**. We compiled species level data for five plant traits: wood density (WD), leaf and stem spinescence, latex production, and leaf size, for

tropical and extra-tropical South and Central American woody species (i.e., the Neotropical biogeographic realm). WD was obtained for 2577 species from ref. [44]. We only used wood density data from Zanne et al.[44], because this study used WD measured in stems, whereas most other studies with available data used WD measured in branches. Leaf size data were obtained for 2660 woody species from Wright et al.[37]. We did not include leaf size from herbaceous species because herbaceous and woody species are influence by different megafauna guilds, suggesting distinct mechanisms, and because this dataset[37] only included data for 253 Neotropical herbaceous species. The presence or absence of stem (and/or branch) spines (mostly thorns, but also prickles) were obtained from Dantas and Pausas[45] for Neotropical savanna and forest species (1004 species) and complemented with other literature sources for other ecoregions (listed in the supplementary materials) using the names of the species for which we had WD and Leaf Size data. Our final stem spines dataset included 2843 woody species. We also compiled data on the presence of latex in plant stems and leaves for all the species for which we had data on other traits (3160 species; references in the supplementary materials). Finally, we also compiled data on leaf spines. While we managed to find leaf spine data for a total of 2173 woody species, we found spinescence in leaves to be especially concentrated in the palm Family (Arecaceae; 198 out of 221 species with leaf spines). Moreover, out of the non-palm species, all but three species also presented stem spines, indicating that, for other taxa, leaf spines might be dependent on the presence of stem spines at the region (in palms, 51% have stem spines). Thus, we only used leaf spinescence data of palm species (694 species) from the global Palm Traits Database 1.0[46].

For wood density and leaf size, we often had more than one trait value per species (1005 and 831 species with more than one trait value, respectively). Thus, we computed the species mean trait value. This rarely occurred for binary traits (spinescence and latex) and, when occurred, the maximum value was used (0 for absence and 1 for presence). This later decision was based in the assumption that omitting the presence of spines or latex is more likely than incorrectly reporting the presence when it is absent. Moreover, some of these traits can be plastic[18].

**From species to ecoregions**. We searched for geographical distribution data (coordinates) from each species-trait dataset (Data available from GBIF using the following doi: WD: https://doi.org/10.15468/dl.3vua3x; Stem spines: https://doi.org/10.15468/dl.ar5ddj; Latex: https://doi.org/10.15468/dl.m8dzjd; Leaf spines: https://doi.org/10.15468/dl.vv8gw4; Leaf size: https://doi.org/10.15468/dl.k98nxc). For this search, we used tools provided by the "rgbif" package for R in which species names are updated to the most recent classification and the returned occurrences also include those associated with synonyms (i.e., the "backbone" method). We labelled the obtained geographical coordinates according to their ecoregion and biogeographical realm (following Dinerstein et al.[47]) and cropped out occurrences falling outside of the Neotropical realm. Since occurrence data was not available to all the species in our initial trait dataset, the number of species used to calculate ecoregion level means was reduced to 2110 species, for wood density, 2133, for leaf size, 2629, for stem spines, 2714, for latex, and 657, for leaf spines. A detailed evaluation of the representativity of this data in relation to ecoregion- and Neotropical- level patterns can be found in the Supplementary Methods. Based on the occurrence data and their ecoregion label, we built a species abundance (columns) by ecoregion (rows) matrix for each trait.

We obtained ecoregion scale abundance-weighted means for continuous traits (WD and Leaf Size) by: (1) Multiplying species abundance in each grid cell of the ecoregion by the mean species value; (2) Summing up the row values; (3) dividing the resulting row sum by the total species abundance (row sum prior to trait multiplication), and (4) calculating the ecoregions' means (across all of the grid cells). For Stem Spines and Latex (binary traits), we used a similar procedure, but the maximum (0 for absence and 1 for presence) value was used instead of the mean in step (1), and step (2) was directly used to calculate the number of presences (i.e., 1 s). Moreover, instead of the steps (3) and (4), we calculated the number of absences as the difference between the total absolute (row sums before trait multiplication) and the values obtained in step (2). This process resulted in weighted means for WD and stem spinescence for 173 ecoregions, and Leaf Size and Latex for 174, out of the 179 Neotropical ecoregions. For leaf spinescence, we used a similar approach, although, because of the fewer species, the abundance estimate from GBIF was less reliable. Thus, we transformed the ecoregion species abundance to presence/absence before multiplying the trait values (0/1 for absence/presence). We obtained leaf spinescence data for 159 out of the 179 Neotropical ecoregions. The species- and ecoregion- level data is provided in the Supplementary Data and in ref. [47].

**Historical megafauna distribution**. We obtained data on historical distribution of megafauna species from the MegaPast2Future/PHYLACINE_1.2 dataset[24], a dataset containing distribution maps (96.5 km of spatial resolution) and functional traits for mammal species of the last 130,000 years. From this dataset, we obtained the probable past distribution of extinct large mammal herbivore (hereafter, "megafauna") species, if these species were still alive today ("Present Natural" scenario; see details below). The "Present Natural" distribution of extinct species in this database is based on the estimated historical distribution (i.e., preceding anthropogenic range modifications) of extant species that are known (from the

fossil record) to have coexisted with the extinct species. In this approach, an extinct species is considered to have been present in a given grid cell if at least 50% of the extant species that were found coexisting with the extinct species in the fossil (and subfossil) record was predicted to have occurred in the same cell prior to anthropogenic range modifications[24,48]. This approach assumes that, since extant and extinct species coexisted in the same locations, they must have had similar ecological requirements. It also assumes that megafauna extinction had anthropogenic causes, instead of causes related to climate change[49], which is largely accepted in the literature[50].

We extracted the "Present Natural" distribution of extinct mammal (coded "EP" for IUCN status; i.e., "extinct in prehistory", meaning before 1500 CE) whose body mass was higher than 50 kg (megafauna), and for which at least 90% of their diet consisted of plants (i.e., strict herbivores). For each Ecoregion, we began by calculating two megafauna-related metrics: extinct megafauna species richness ($M_{rich}$) and their mean body mass ($M_{bm}$). For this, we cropped the distribution maps of the megafauna species (containing 1 for presence and 0 for absence of each species) to the Neotropical realm. To calculate $M_{rich}$, we (1) counted species presences within each of the grid cells in the global grid (i.e., calculated the cell's megafauna richness); (2) assigned the corresponding ecoregion label to the resulting richness grid cells, subset the richness cell values corresponding to the Neotropical region; and (3) calculated the mean for each Neotropical ecoregion. For $M_{bm}$, we replaced the presences of the megafauna species in the initial raster object (grid cell map of each megafauna species) by their body masses and calculated the grid cell-level mean body mass, before calculating the ecoregion-level means. We also calculated megafauna density and secondary productivity based on allometric equations that relate these metrics to megafauna body mass. However, we did not used megafauna density and secondary productivity because they were strongly correlated to megafauna richness (Supplementary Fig. 3). More details on how these metrics were calculated can be found in the Supplementary Methods.

We also obtained diet preference information from the literature for most megafauna species that occurred in the Neotropical region (details and references in the Supplementary Material). Based on these information, we calculated the richness of large browser ($MB_{rich}$ for megabrowser richness), grazer ($MG_{rich}$ for megagrazer richness), and mixed-feeder ($MMf_{rich}$ for mega mixed-feeder richness) species by sub setting the megafauna species by grid cell array before the richness calculation in order to select only species that were classified within the correspondent subgroup.

**Extant herbivore mammal distribution**. We also compiled data on the distribution, body mass and diet of extant and recently extinct (i.e., extinct after 1500 CE) herbivore mammal species (for simplicity, called 'extant' species in this study). As with megafauna maps, the distributions used represented reconstructions for periods preceding anthropogenic reduction of extant herbivores ranges ("Present Natural" scenario), based on abiotic, biotic and geographic variables[48], rather than the currently observed distribution. This scenario was used because modern anthropogenic range reductions are too recent to produce substantial geographic effects at this spatial scale. These data were obtained by sub setting the Mega-Past2Future/PHYLACINE_1.2 dataset to exclude species that were coded "EP" for IUCN status and that were not strict herbivores (at least 90% of the diet constituting of plants). We subsequently associated diet information to these species using data from ref. [51] and excluded all species that did not feed mainly on aboveground vegetative plant tissues (i.e., species that fed mostly on fruits, seed, roots were excluded). This later filtering was because the number of herbivores that feed mostly on seed and fruit increase with decreasing size (and this dataset included small mammals). We subsequently calculated the same metrics as for the extinct megafauna species (except for the richness of mixed-feeders as our source for diets[50] labelled species according to dominant feeding pattern). For this, we used the same approach described for extinct megafauna species. We did not use a size threshold for extant species because there were only 13 extant mammal herbivore species with over 50 kg in the Neotropical region, most of which were grazers (9 species; 4 species were mixed-feeders and none were browsers). Therefore, we relied on the mean body mass metric calculated for extant mammals to detect potential size-related effects.

**Climate, soil, fire, insularity, and hurricanes**. For each Ecoregion, we obtained data on climate (mean annual precipitation and temperature, and rainfall seasonality) and soil (sand content, pH, and cation exchange capacity) variables. Climate data was obtained from WorldClim 2.1 (10 min spatial resolution) and was based on climate data from 1970 to 2000[52]. Soil data were obtained from SoilGrids (5 km of spatial resolution)[53], and consisted of mean values for two depths, 0.05 and 2 m. We calculated Ecoregion level means for all of the soil and climate variables after intersecting the climate and soil grid maps with the ecoregion map.

We obtained the number (a proxy for frequency) and intensity of wildfires per ecoregion using the MODIS active fire location product (MCD14ML)[54]. We only considered fires (i.e., hotspots) with detection confidence of 95% or higher occurring from November 2000 to December 2019 (both included). To ensure that only wildfires were considered, we associated each fire pixel with a land cover type (300 m of spatial resolution) from ref. [55] for a buffer area of 1000 m surrounding the fire pixel centroid. We excluded all of the fires occurring in areas in which more than 10% of the surrounding land cover pixels corresponded to agricultural, urban

and water classes. We calculated the number of wildfires per ecoregion area by dividing the fire count of each Ecoregion by the ecoregion area, and multiplying the resulting value by the proportion of vegetated land cover pixels (same classes used to exclude fires in anthropogenic areas and water bodies above). Fire intensity was estimated as the average fire radiative power across all detected MODIS hotspots in the ecoregion. Ecoregions lacking large preserved vegetated areas (criteria above) were excluded from subsequent analyses.

Using the ecoregion map, we also classified ecoregions into insular (1), when most of the ecoregion area was located in islands, vs. continental (0), otherwise. This was performed because island biogeography theory predicts that, in island, species richness should be low due to low colonization and high extinction rates. Insularity has also been shown to reduce megafauna body size (i.e., the island rule), even though the mechanisms are not fully understood[56]. We also compiled data on hurricane activity, as woody density was suggested to confer resistance against this disturbance[57]. We used data from 1990 to 2019 from the HURDAT2 dataset[58], containing six-hourly information about the location of all of the known tropical and subtropical cyclones (0.1° latitude/longitude). We used the sum of hurricane occurrences per ecoregions divided by ecoregion area as an indicator of hurricane activity.

**Statistical analyses**. To understand megafauna patterns, we began by fitting (multiple) regression models with habitat-related (fire, climate, soil) and geographical (insularity) variables as predictors. We expected that megafauna richness in general was higher under savanna conditions (arid nutrient-rich or mesic nutrient-poor environments with frequent fires)[1,22]. We also expected that megafauna richness and body mass were affected negatively by insularity (i.e., following the island biogeography theory and island rule). Before the analyses, we tested the correlations among all of the variables that would eventually be entered as predictors in the same model for both the megafauna and trait models (Supplementary Table 1), in order to avoid multicollinearity associated with highly correlated variables (here, $r \geq 0.60$). Since mean annual precipitation and soil pH were strongly positively correlated ($r = -0.78$), for all of the analyses (including the analyses with functional traits, described below), model selection was performed separately for these two variables (i.e., two different model selection procedures, one containing each of the two variables among the initial set of predictors). We selected the best among the two resulting models as that with the lowest AIC (differences higher than two points in all of the cases). To make sure that no multicollinearity remained we also calculated the Variation Inflation Factor (VIF) for all of the predictor variables as 1/tolerance, where tolerance is calculated as 1 minus the $R^2$ of all of the model regressing a predictive variable against all of the other predictors. In all of the models, VIF was 3.33 or smaller (i.e., a tolerance of 0.30 or higher), indicating absence of multicollinearity.

Model simplification was carried interactively using stepwise (both forward and backward) searching for the model with the lowest AIC (using R's "step" function) and subsequently retaining only the significant variables ($p \leq 0.05$). We calculated the Pearson r statistics as a measure of effect size for the selected variables as well as the associated confidence intervals, using the packages "parameters" and "effectsize" for R. The average contribution of each predictor variable was also calculated, using the package "dominanceanalysis", as the mean difference in $R^2$ before and after removing the target variable from models containing all of the possible subset combinations of the selected predictor variables, including the full selected model.

For testing whether the studied plant functional traits were related to our megafauna indicators, we fit linear models to WD and leaf size, and generalized linear models (GLM; binomial family) for spinescence and latescence, using ecoregion as the unit. For spinescence and latescence, we used the matrix containing the count of spiny/latex and non-spiny/non-latex plants (species abundance; for stem spines and latex) or number of plants with or without spines (for leaf spines; see above) as response variables. The predictor variables included the animal indicators for extinct megafauna and extant herbivores, as well as climate, soil, and fire predictors (and, for WD, hurricane counts). Because total, as well as megagrazer, megabrowser, and mega mixed-feeder species richness were strongly positively correlated (Supplementary Table 1), we used the richness difference between grazers and browsers to evaluate the effect of diet (Supplementary Fig. 1). For consistency, we used the same diet variable for extant and extinct species. Since we did not identify strong correlations among extinct megafauna and extant herbivore indicators (Supplementary Table 1), these variables were all entered simultaneously in the same initial models. As with the analyses of the megafauna indices, we also used r as effect size and calculated the average predictor contribution in terms of $R^2$ for these models. For the later, we used the MacFadden Pseudo-$R^2$ in the GLM models as implemented in the "pscl" and "dominanceanalysis" packages for R, as this statistic is the most comparable with $R^2$ from linear multiple regression (Maximum Likelihood and Cragg and Uhler's Pseudo-$R^2$ were also calculated for the logistic models), and adjusted $R^2$ for continuous traits. Islands were not included in these models, as island plants were expected to respond differently due to the effects of insularity on animal species richness, precluding megafauna and extant mammal richness from being accurate proxies for consumer abundance. For stem spines, we always included a quadratic term to both megafauna and extant mammal herbivore body mass, as evidence suggest that medium-size herbivores (i.e., approximately 250 kg) are important

selective drivers of this trait[12]. If a significant relationship with our herbivory indicators (both extant and extinct) were significant but not indicative of a selective effect by herbivores (for more defended plants), this relationship was discarded (along with related variables, such as diet); this happened only once, for leaf size, which increased with extant herbivore richness (Supplementary Table 8).

For all of the general linear regression models, assumptions of normality, homoscesticity and lack of spatial autocorrelation in the residuals were checked using the Kolmogorov–Smirnov, Breusch–Pagan and Moran's I tests, respectively. For the later, ecoregions were considered neighbours when they were adjacent and non-neighbour otherwise. In some cases, heteroscesticity was detected and, thus, the significance of the coefficients was tested using heteroskedasticity-consistent covariance matrix estimation. If one or more variable lost their significance they were stepwise removed from the final model, beginning by the least significant, until all remaining variables had a significant effect. Overdispersion in the generalized linear model was also detected and dealt with using overdispersed binomial logit models, as implemented in the "dispmod" package for R, in which weights are interactively calculated and used to maintain the residual deviance lower than the degrees of freedom. To confirm that the detected associations between megafauna indices and plant traits were robust, we also tested the coefficient significance using randomization of the plant species by ecoregion matrices (see Supplementary Methods for details).

To test the prediction that Neotropical ecoregions could be broadly classified into the three hypothesised antiherbiomes, we used hierarchical clustering on principal component axes of the ecoregion by trait matrix (five plant traits, standardized to zero mean and unit variance). We selected the number of clusters associated with the highest loss of inertia (within group variability) when progressively increasing the number of clusters, using the R package "FactoMineR". This procedure allowed the recognition of large regions characterised by specific patterns of defence strategies ('antiherbiomes'). We subsequently tested for axes score, megafauna and environmental differences among the resulting antiherbiomes to verify whether and how trait, climate and soil patterns matched those described for African ecosystems, and to understand the megafaunal differences among the antiherbiomes. For these comparisons, we used Kruskal-Wallis and post-hoc pairwise Dunn tests, using the Benjamini & Hochberg[59] (1995) correction of P-values for multiple comparisons in both cases, and exclusively included continental ecoregions. For spines, we used the proportion of spinescent plants/species (rather than the number of "yes" and "no" used on previous analyses) in the principal component analysis. Because palms were missing from 20 ecoregions, we completed the values for these ecoregions using predicted model probabilities. To better understand these associations between traits and the environmental and megafauna variables, we also regressed the PCA axes against the same predictors used for traits.

We also developed a framework to identify forest ecoregions most likely to have experienced a biome shift after megafauna extinction using antiherbiome, biome and megafauna distribution data. Ecoregions likely to have experienced a savanna-to-forest shift since the Pleistocene are those that: (1) are currently forest-dominated; (2) are classified in antiherbiomes analogous to African arid nutrient-rich or mesic nutrient-poor savannas; and (3) were megafauna- and, especially, megagrazer- rich during the Pleistocene (richness equal or greater than the 0.75 quantile: 14 species for $M_{rich}$, and 3 for exclusively grazing species; $MG_{rich}$). We validated the distribution of these areas with fossil evidence (22 sites) from the Last Glacial Maximum and mid-Holocene (see Supplementary Methods and Supplementary Table 2). For this, we also used information about the present dominant vegetation type in the fossil sites, extracted from the reference sources (see Supplementary Table 9), to segregate savanna-forest shifts from data coming from stable savanna patches within forest or long-term savanna regions. We also contrasted the predicted patterns with the present location of savanna patches within the Amazon Forest region from ref. [60].

All statistical analyses and data handling were carried out in the R (v.4.0.2) environment, using the previously mentioned packages, in addition to FSA, gridExtra, grid, lattice, lmtest, latticeExtra, olsrr, raster, rgdal, rgeos, sandwich, spatialreg, spdep and vegan, using codes provided in ref. [47].

**Reporting Summary**. Further information on research design is available in the Nature Research Reporting Summary linked to this article.

## Data availability statement

This study is based on open source data compiled from the literature or downloaded published datasets, such as: MegaPast2Future/PHYLACINE_1.2 (https://doi.org/10.5281/zenodo.3690867), Diet preferences in terrestrial mammals worldwide (https://doi.org/10.5061/dryad.6cd0v), WorldClim2 (http://www.worldclim.com/version2), SoilGrids (https://www.isric.org/explore/soilgrids), MODIS active fire location product (https://modis-fire.umd.edu/af.html), HURDAT2 (https://www.nhc.noaa.gov/data/#hurdat), Global wood density database (http://hdl.handle.net/10255/dryad.235), PalmTraits 1.0 (https://doi.org/10.5061/dryad.ts45225), Dantas and Pausas, 2020 (https://doi.org/10.5061/dryad.3xsj3txc0), Flora do Brasil (http://floradobrasil.jbrj.gov.br), GBIF (www.gbif.org) and Ecoregions2017 (https://ecoregions.appspot.com/). The curated data generated in this study have been deposited in the Zenodo database (https://doi.org/10.5281/zenodo.5752131) and was also added as Supplementary Data alongside this

article. A Source Data table containing the PCA scores used for antiherbiome contrasts is also provided alongside this article.

## Code availability statement

All codes used in the main analyses and figures have been deposited in Zenodo (https://doi.org/10.5281/zenodo.5752131).

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

## Author contributions

V.L.D. conceived the idea, with inputs from J.G.P. V.L.D. compiled and analyzed the data, and wrote the first draft of the manuscript. J.G.P. critically contributed to revisions and authorized submission.

## Competing interests

The authors declare no competing interests.
