## [Peer Review File · Nature Communications]

Reviewers' Comments:

Reviewer #1:

Remarks to the Author:

Dantas and Pausas study the effect of past megaherbivores (megafauna) on the present-day distribution and abundance of defence traits (e.g. spines, thorns) that may have evolved as adaptations to deter megaherbivores in the past, focusing on Latin America (that lost most/all of its megaherbivores). They define 'antiherbivomes' as areas/ecoregions that have been strongly influenced by past megagrazers/browsers, as well as in interaction with climate and disturbance variables (e.g. fire). The authors make an analogy to African savanna types that are still strongly influenced by megafauna (e.g. elephants). The authors also show that certain bioregions that currently lack megaherbivores (e.g. forest patches), were probably influenced by megafauna in the past, and have likely transitioned from savanna into forest. The central question and approach is interesting, and very relevant as it increasingly shows how current biodiversity patterns and traits are influenced by past mechanisms and processes (e.g. Quaternary climate change), but the focus on past biotic interactions is rare. As such, this is an interesting and novel contribution to the field. However, I do have concerns about data quality and the analytical approach, and how sensitive the results and conclusions are to these (details indicated below). With such correlative approaches, it is very important to be careful about the underlying data and approach, and also include discussion on alternatives that may explain emerging patterns.

Major comments:

Data for plants: the underlying species-level data for traits, on which the study relies, is relatively poor given that conclusions are drawn for the Central and South-American floras, but far less than 1% of the species are included (if I'm not mistaken). In fact, for leaf spines, the authors only focus on palms (Arecaceae), which is not representative for the savanna flora, as it is a typical rain forest family. I'm therefore concerned to what extent these data reflect general trait distributions across the flora. As I have collected these types of trait data myself, I know that a much better job can be done, including more species and trait data on several of the defence traits investigated (leaf area, wood density, leaf and stem spines), which would also improve the number of species for which trait data would be complete (which is not essential as statistics are performed for each trait separately, but it would improve our understanding of trait relationships in the different ecoregions, and their relation to environment). For example, local floras could be assessed for defence traits, herbaria for leaf size.

Data for mammals: the data used for past megaherbivores relies on the co-occurrence with extant animals in the fossil record. However, there is another issue not presented, which is how present distributions of extant species (and thus past species) are inferred, called 'range modifications' in the Phylacine data. This should be acknowledged, as well as the limitations, and how it affects the conclusions (which modification is used, and why). In addition, the authors define megafauna as >50 kg - why? This does not follow the existing definitions (>40 or 44kg, or >1000 kg - especially when it comes to savannas the >1000kg may be a better proxy, similar to elephants), and it would be essential to test how this cut-off value affects the results. How do results change when choosing >1000 kg? Or >40 kg? Or >60 kg? Do they hold in a similar way? Also, the authors compare results of past megafauna, to the effect of current herbivores on plant traits (finding no significant effect, and thus concluding that present-day traits are largely the result of past interactions). However, for present-day herbivores, the authors do not focus only on megafauna. So datasets differ in two ways (extinct vs. present natural; megafauna only vs. all), instead of one. I suggest also focusing on megafauna only for this 'sensitivity' analysis, to make them comparable.

Averaging for bioregions: first, a bit more detail on the bioregion classification needs to be provided, as this is the unit of analysis, and there is a lot of different opinions about this classification. How sensitive are results to this classification? Second, why use the ca. 60 (or so?) bioregions as unit in the analysis, why not use the grid cells the data comes from directly? This would increase the sample size, avoids making arbitrary averages, and the interpretation would not rely on 'bioregion'. I also wonder whether a cut-off value was used to include the unit in the analysis, for example, it should have data for at least 100 plant species occurring in there? Or at

least 10? Because if it's based on 1 species, it's clearly biased...

I would be interested to see the raw data for these analyses, the plant-species matrix by traits, as well as the species x bioregion dataset for both plants and mammals. Please submit this if invited for a resubmission/revision.

Data analysis: the study uses simple linear and generalized linear models to associate plant trait averages in bioregions, to mammal richness, climate and disturbance (fire etc.) variables. It uses step-wise model selection, and then assesses the relative importance of the variables. It is an ok approach, but a much more elegant approach would be to use structural equation models, especially because the authors talk about potential direct and indirect effects, which could be assessed in these models (as well as different response variables, so all traits could be evaluated in a single analysis). Regardless on whether SEMs are used, it would be important to have a distance measure in the model to correct for potential spatial autocorrelation (or evaluate with a SAR model).

Data analysis – sensitivity analyses/simulations: Second, as mentioned, the study is correlative, and correlations could arise because of potential other (unmeasured) factors. It would therefore be essential to do several sensitivity analyses. For example, when randomising trait averages (the response) across bioregions 1000x times, does the observed standardized effect come out as unique (outside 95% probability of simulated datasets?) And when using a trait as response that is not supposed to relate to past megafauna, is megafauna indeed not significant? For examples, see here: https://royalsocietypublishing.org/doi/full/10.1098/rspb.2019.2731?url_ver=Z39.88-2003&rfr_id=ori:rid:crossref.org&rfr_dat=cr_pub=pubmed

The discussion feels a bit 'light' and could do with more thoroughly relating results and interpretation to the existing literature, and especially make a more in depth comparison to Africa (and Asia?).

Minor comments:

Use more informative headings in Supplementary information tables and figures, explain for figures what can be seen and some sort of interpretation, and explain ALL abbreviations in tables. Also in the main text some abbreviations in figures are not explained in caption. Abbreviations are generally quite confusing, and it would be more helpful to avoid them where possible.

Table S3 MAR ◊ MAP

Why are abiotic variables often not significantly explaining defence traits in the linear and generalized linear models, but then are included when defining the 'antitherbiomes' based on cluster analyses? Are they suddenly correlated to them in the cluster analysis, and why? Perhaps explain better.

"There are multiple mechanisms by which plants defend themselves from large herbivores 5,7,11–13, and these mechanisms differ with climate and availability of soil resources" – please describe in more detail how they interact.

"mesic nutrient-poor savannas dominated by broad-leaved plants that mainly rely on leaf defences" – normally, low nutrients lead to small leaves and low specific leaf areas (sclerophylly) – this is not the case here? Why not? This contrasts with several other low-nutrient biomes and could be briefly discussed.

"A third ecosystem type can be readily identified in tropical Africa, that is, forests, in which low levels of megafauna herbivory and high productivity enable quick canopy escape, and, therefore, plants are largely undefended" – hm, not sure if I believe this, many rainforest plants have hooks and spines, often also related to climbing. Especially in palms (one of the clades focused on here...)

"large mammal herbivores over 50 Kg that became extinct in pre-historical times" – define what pre-historic times are.

"Woody density was especially high and leaf size especially low where extinct large grazer (i.e. megagrazers) richness was greater than that of large browsers (i.e. megabrowsers; Fig. 2; Table S3, S5)." – at this point in the text it is not clear why distinguishing grazers and browsers, clarify here or before.

"In contrast, extant mammal herbivores did not explain any proportion of the variability observed in antiherbivory defence traits (Fig. 2; Table S5)." – I do not see this in Table S5 (This seems to only relate to using megafauna as response? It would be good to have the comparison.) – not clear which tables relate to which analyses.

"the relationship with extant herbivore richness indicated bottom up control of these traits on modern mammal distribution (i.e. a significant negative relationship; Table S7)." – hard to follow, please explain.

"A cluster analysis on the trait by ecoregion matrix (see Methods for details) supported the hypothesis that three distinct antiherbiomes can be recognized in the Neotropics." – bit more details on the analysis needed (without having to read materials/methods first)

"(3) megafauna sensitive assemblages" – confusing, what does 'sensitive' mean or refer to?

"are not under strong selection during the Holocene"- Why not? They are still there, and correlate with past megafauna?

"where grazers were relatively more abundant (Fig. 2)" - Compared to browsers, make clear

"suggesting that this defence trait is especially adaptive under grazer-controlled fire regimes." – why fire?

"In Africa, large browsers and mid-size social mixed-feeder species are important predictors of stem spines, although the latter are presumably more important 12. In our study, including mixed-feeder did not improve model fit (not shown)." – please do show

"that leaf size is especially useful as a defence mechanism in nutrient-rich soils and drier climates." – please repeat why

"For many species we had more than one trait value, so we computed the species trait value as the mean for quantitative traits" – for how many?

"We obtained geographical distribution data (coordinates) from GBIF for all of the species in each species-trait dataset" – Cleaned? How dealaed with biases? Taxonomical issues?

"Because palms were missing from 20 ecoregions, we completed the values for these ecoregions using predicted model probabilities." Bit tricky when based on palms only...

Renske Onstein

Reviewer #2:

Remarks to the Author:

The paper proposes an interesting analysis of the effect of large mammal herbivores on the evolution of plant traits and their biogeography in Central and South America. The authors show the presence of 3 distinct regions or "antiherbiomes" which are characterized by similar plant defense strategies, megafauna history, and environmental factors. They also identify regions which have shifted from being grass-dominated to forest-dominated. Their results suggest that herbivorous megafauna had a significant and long-lasting effect on plant traits and biome distribution.

The topic is both interesting and relevant to the increasing focus on the study of the ecosystemic role of large terrestrial mammals. In this context, the research questions addressed by the authors are of great importance not only for plant-animal ecology but also for other related fields such as evolution and biogeographic modeling. The methodology makes good use of a lot of different data sets and datatypes by including trait data of both animals and plants. However, I think there are some key points to be addressed or clarified to improve the paper. The statistical approach is current and well designed. The figures are nice and the results well presented also in the supplementary. The discussion section is also interesting and well written but I think there are a few important points that are overlooked.

One of my main concerns is that the authors relate the effect of megafauna to plant traits based on correlations between species richness and mean body mass. However, herbivore population density is probably just as important than species richness or mean body mass because population density determines the intensity of "animal disturbance" on the landscape and on plants. For example, during the Pleistocene in the tundra-steppe ecosystem the species richness was low but the density of mammoths and other large herbivores was probably quite high and thus it had a significant effect on the ecosystem. In tropical grasslands the herbivore biomass should be higher than tropical forests. Similarly, for mean body mass, a higher mean body mass doesn't necessarily relate to a higher animal biomass per area. The relationship between body mass and density can vary considerably. Considering that herbivore density is a key parameter in plant-animal interactions I wonder how the absence of such factor might influence the authors' results. Another unclear point is that even though herbivores have been divided in grazers, browsers, and mixed-feeders, there can be substantial differences intra- and inter-species depending on the ecosystem. Extinct megafauna in the Neotropics probably consumed large amounts of fruit in forested areas. In those cases, plants probably invested in other key traits or defenses. In this regard, the authors should clarify their definition of herbivores because in the Phylacine database herbivores are not differentiated according to their plant dietary preferences (fruits, seeds, leaves). I do not see how including all these different herbivore guilds is relevant to the plant traits used in the analysis.

I don't completely agree with the authors discussion in regards to leaf size and wood density. Wood density is related to many eco-physiological properties and has high intra-specific variation. Authors conclude that wood density is higher where grazers were relatively more abundant but the wood density patterns presented in Chave et al. 2009 do not confirm this trend. There is also a trade-off between growth rate and wood density. Lower wood density is associated with increased growth rates and potentially provides a faster escape from the herbivore trap at the cost of reduced mechanical strength. In regards to leaf size, Wright et al. 2017 state that differences in leaf to air temperature are key to shaping leaf size, I think this should also be discussed. Also, the Wright et al. included a large range of growth forms, so direct comparisons might be tricky. In general, I find it somewhat confusing that only woody species are discussed because grazers and mixed-feeders are associated with feeding on herbaceous vegetation and thus the antiherbivores are only described in terms of woody species traits but grasses also develop herbivore defenses.

I think there should also be a paragraph to discuss the interpretation of these results in term of short-term and long-term adaptations. As mentioned by the authors in regards to spines, some of the plant traits included in the analysis can be very plastic and the lack of phylogeny correction in the analysis does not allow to fully evaluate the results in the evolutionary context. For a better evaluation of the evolutionary effect phylogeny correction should be included in the analysis.

A few minor comments:

- From what I understand the term "Neotropical" is used throughout the paper when referring to South and Central Americas. I find this confusing because the terms Neotropics and Afrotropics are often used in tropical ecological literature when referring strictly to tropical biomes. I'm not sure why the term Neotropical is used since this paper covers South America which includes a wide range of biomes.
- See annotated comments in text
- Diet studies of extinct herbivores often relate C3 plants to a closed-habitat browser diet and C4 to a grazer diet, however there can also be a mix of C3/C4 plants in the diet making difficult to assess. I would add a bit more information in your methodology on how you determined grazing

and browsing for extinct animals.

- The legend of figure 5 should be made more clear
- Please discuss more why the Amazon basin, the cerrado and the Atlantic Forest are under LDW and the Western Amazon is not LDW.

Some useful references

1. Berzaghi F, Verbeeck H, Nielsen MR, Doughty CE, Bretagnolle F, Marchetti M, et al. Assessing the role of megafauna in tropical forest ecosystems and biogeochemical cycles – the potential of vegetation models. *Ecography*. 2018;41(0):1–21.
2. Chave J, Coomes D, Jansen S, Lewis SL, Swenson NG, Zanne AE. Towards a worldwide wood economics spectrum. *Ecology Letters*. 2009 Apr 1;12(4):351–66.
3. Hatton IA, McCann KS, Fryxell JM, Davies TJ, Smerlak M, Sinclair ARE, et al. The predator-prey power law: Biomass scaling across terrestrial and aquatic biomes. *Science*. 2015 Sep 4;349(6252):aac6284.
4. Santini L, Isaac NJB, Maiorano L, Ficetola GF, Huijbregts MAJ, Carbone C, et al. Global drivers of population density in terrestrial vertebrates. *Global Ecology and Biogeography*. 2018;27(8):968–79.
5. Vizcaíno. (PDF) Evaluating Habitats and Feeding Habits Through Ecomorphological Features in Glyptodonts (Mammalia, Xenarthra). ResearchGate [Internet]. 2011 [cited 2018 Dec 5]; Available from: https://www.researchgate.net/publication/257931077_Evaluating_Habitats_and_Feeding_Habits_Through_Ecomorphological_Features_in_Glyptodonts_Mammalia_Xenarthra
6. Zhu D, Ciais P, Chang J, Krinner G, Peng S, Viovy N, et al. The large mean body size of mammalian herbivores explains the productivity paradox during the Last Glacial Maximum. *Nature Ecology & Evolution*. 2018 Apr;2(4):640–9.
7. Doughty CE, Wolf A, Morueta-Holme N, Jørgensen PM, Sandel B, Violle C, et al. Megafauna extinction, tree species range reduction, and carbon storage in Amazonian forests. *Ecography*. 2016 Feb 1;39(2):194–203.

Reviewer #3:

Remarks to the Author:

This paper presents a new analysis looking for evidence that the geographical some functional traits (plant defence traits) of South American flora can be largely explained by the presence of past megafauna. The approach taken is to combine (i) information on the spatial distribution of plant species (ii) data on species defence traits (spinescence, leaf size and wood density) and (iii) information on past distribution of megafauna (previously published by other authors) that infers distribution based on coexistence with extant fauna in the fossil record, and then using yeh current distribution of extant fauna.

The emerging insights are that (i) past megafauna can explain a substantial fraction of modern plant defence trait distribution(i) it is possible to identify three broad “antiherbiomes” in South America” based on plant traits, analogous to ones identified in Africa where megafauna are present. They also suggest there are regions that were grassy under megafauna presence but shifted to forest domination following megafauna extinction.

The novelty of this work is a clear quantification of the proposition that the extant flora of the Americas carry the “ghost” signatures of recently extinct megafauna, which enables both an interpretation of why these traits are found where they are, and opens up an avenue to refine understanding of the past ecology and distribution of the megafauna, and how South America’s ecosystems changes following megafaunal extinction. These topics have been the subject of much speculation and this paper would make a substantive analytical advance in the field. The paper would also influence our understanding of South American ecology, which tends to ignore the possible role of past megafauna in shaping modern day ecosystems.

Overall, I am intrigued by this paper and its argument, but feel it relies rather heavily on broad

spatial statistical analysis, without critical interrogation of the datasets involved or the sweeping conclusions buried in the statistics and tables. Figures 1 and 2 illustrate this. A key part of the argument is here and warrants greater exploration. The most striking result is that most of the biomes of Brazil carry the signatures of the LDW antiherbivore (Fig 2). Looking more closely, this seems to be the conclusion derived from high levels of leaf spines in palms in this region, and higher wood density (Fig. 1). The prevailing ecological interpretation has previously been that high wood density is linked to soil properties in the crystalline shield region (deep soils, low fertility) compared to high fertility and shallow soils in the Amazon lowlands (see papers by Beto Quesada et al for Amazonia). I wonder if the soil variables picked here really capture this gradient, whether past megafauna richness really is the predominant driver of patterns in wood density across the Neotropics? I may be wrong, but I feel this requires more sceptical exploration, beyond simple reporting of broad spatially correlated statistics. How much do we trust the inferences of the PHYACINE database? Have the soil environmental variables been accurately captured? (no reference to the extensive literature on this)?

Is Figure 2 really saying that much of Amazonia was a herbivore accessible mosaic in the Pleistocene? Or is it that sub-canopy farms had some herbivory pressure in what was still extensive closed-canopy forest?

Lines 52-56: It would be helpful to provide some mechanistic speculation/explanation on why these different environmental conditions lead to these antiherbivore traits.

Also, I am curious where dry, nutrient-poor savanna or mesic, nutrient-rich savanna fit into this antiherbivore framework?

Line 66 While spinescence is an obvious plant defence trait and leaf size corresponds to your antiherbivores, please justify why wood density would might be associated with plant defence strategies

Line 68 The use of the prehistoric data needs some explanation and critical consideration. On what assumptions is it based, and how reliable is it

Line 72: I recommend use of "humans" rather than "men" here and throughout

Line 81 onwards. It is important that use of body mass is clarified at the start. Mean body mass means mean body mass of megafauna species - it does not (and cannot) account for relative abundance of the species. The authors know this but with our clear explanation it may escape the general reader.

Line 169 onward - are all paleoclimate models consistent in predicting which areas where savanna in the Pleistocene? Is there any supporting palynological evidence?

Line 480 - More information is needed on the fossil data used to identify the forest areas that were formerly savanna. Can a table of the sites and evidence be provided in the references? I would like to be convinced these were not small patches of savanna in a forest biome (as some are now), or where there is evidence of a shift from savanna-forest mosaic to continuous closed-canopy forest following megafaunal loss. This is a key part of the argument but is rather rushed in the paper

The order in which figures in supplementary material are mentioned does not match their sequence of presentation

Figure 5 This is potentially a very important figure but I am confused. Five points indict points that

are non forest today, but the salmon points indicate Amazonian savanna patches (and hence also non-forest). If we assume these savanna patches are persistent savanna (perhaps for edaphic reasons) the fossil evidence for grassland to forest transition in Amazonia is limited to periphery of the modern forest biome. The evidence-based on the antiherbiome is still there perhaps but this needs some careful discussion. The evidence in the Atlantic forest and caatinga is perhaps stronger.

Figure S1 needs a better caption

Table S6, S7 etc Explain the table variables in every caption please

REVIEWER COMMENTS

Reviewer #1 (Remarks to the Author):

Dantas and Pausas study the effect of past megaherbivores (megafauna) on the present-day distribution and abundance of defence traits (e.g. spines, thorns) that may have evolved as adaptations to deter megaherbivores in the past, focusing on Latin America (that lost most/all of its megaherbivores). They define ‘antiherbiomes’ as areas/ecoregions that have been strongly influenced by past megagrazers/browsers, as well as in interaction with climate and disturbance variables (e.g. fire). The authors make an analogy to African savanna types that are still strongly influenced by megafauna (e.g. elephants). The authors also show that certain bioregions that currently lack megaherbivores (e.g. forest patches), were probably influenced by megafauna in the past, and have likely transitioned from savanna into forest. The central question and approach is interesting, and very relevant as it increasingly shows how current biodiversity patterns and traits are influenced by past mechanisms and processes (e.g. Quaternary climate change), but the focus on past biotic interactions is rare. As such, this is an interesting and novel contribution to the field. However, I do have concerns about data quality and the analytical approach, and how sensitive the results and conclusions are to these (details indicated below). With such correlative approaches, it is very important to be careful about the underlying data and approach, and also include discussion on alternatives that may explain emerging patterns.

R: Thank you for your useful and insightful comments and suggestions, which certainly helped to improve the quality, clarity and reliability of our approach and results. We made an enormous effort to address all yours and other reviewers’ main concerns and hope that you are happy with the improvements.

Major comments:

Data for plants: the underlying species-level data for traits, on which the study relies, is relatively poor given that conclusions are drawn for the Central and South-American floras, but far less than 1% of the species are included (if I’m not mistaken). In fact, for leaf spines, the authors only focus on palms (Arecaceae), which is not representative for the savanna flora, as it is a typical rain forest family. I’m therefore concerned to what extent these data reflect general trait distributions across the flora. As I have collected these types of trait data myself, I know that a much better job can be done, including more species and trait data on several of the defence traits investigated (leaf area, wood density, leaf and stem spines), which would also improve the number of species for which trait data would be complete (which is not essential as statistics are performed for each trait separately, but it would improve our understanding of trait relationships in the different ecoregions, and their relation to environment). For example, local floras could be assessed for defence traits, herbaria for leaf size.

R: While we made an extra effort to expand our dataset, as will be explained with detail later in this reply, first, we would like to present a different perspective on how representative our former dataset actually was in relation to the flora.

First, in this type of study, total number of species in relation to the Neotropical flora cannot be considered alone as a direct indicator of whether the trait data “reflect general trait distributions across the flora”. Many species are likely to be rare and contribute little to ecoregion level patterns. A more important aspect is whether the used species and their distributions are or are not representative of the underlying pattern and processes within and across ecoregions. Depending on how species are sampled, even a very small number of species could explain a large variability of the total ecoregion trait, if these species are the dominant ones. While we do not have detailed data on species dominance, dominant species are much more likely to be well represented in the GBIF database, whereas rare species are more likely to be absent and, therefore, to be excluded due to lack of data, as we move from species to ecoregions.

One straightforward approach is comparing previous accounts of species richness across ecoregions with those observed in our datasets. Here we use data from (Kier *et al.* 2005) reporting global patterns of species richness across ecoregions to evaluate the extent to which the species richness in our dataset were correlated to values reported by them. 135-136 ecoregions in our data was also represented in (Kier *et al.* 2005); out of the 173-174 (152 for leaf spines) for which we had trait data. For all four traits (i.e., here including Leaf Spines), the species richness of the ecoregions was strongly correlated with estimates from (Kier *et al.* 2005), with correlation coefficients ranging from 0.62, for stem spines, to 0.77, for leaf size (and was 0.75 and 0.71 for leaf spines and WD, respectively). This strongly suggest that our data fairly represent the patterns of the ecoregions. If we compared the proportion of the ecoregions’ floras that was represented in our dataset (Kier *et al.* 2005), we will see that, except for leaf spines (which is based on palm data only and will be discussed later), the assemblages in our former data represented 9-11 %, on average, of the vascular plant flora within these ecoregions (for stem spines: 9%; wood density: 11%; and leaf size: 11%). However, one should notice that our study only concerns woody species. Thus, it would be more appropriate to compare our data only with the woody plant flora, rather than the total vascular plant flora. Estimates from Engemann *et al.* (2016) indicate that woody species comprise approximately 39 % of the total richness of vascular plant species in the Americas. If we applied this proportion to the flora in (Kier *et al.* 2005), our species would represent 22-29%, on average, of the total woody flora of the ecoregions and only 3-5 of the ecoregions would be represented by less than 3 % of its woody flora. These results are now reported in the Supplementary Methods and mentioned in the discussion (see the paragraph before the last). However, the results that are presented in the manuscript are those concerning our new dataset (rather than the ones above; see below).

While the information given above is, in our opinion, more informative of how representative our data was in relation to the regional patterns, we also evaluated how the total species richness across the entire Neotropical region compares with that in our dataset. Even when the flora is considered in absolute terms, our data represented much more than 1% of the woody flora for most traits (except leaf spines, that will be

discussed later in this reply). According to estimations by (Govaerts 2001), the flora of the Neotropical region comprises 115,242 species of seed plants. Applying the percentage of 39 % (percentage of woody species; see above paragraphs) would result in 44,944 woody species. Our former dataset (approx. 2500 species per trait), therefore, would represent 6 % of the total flora. Considering our new dataset (see below), this number increased, but in the previous versions we failed to report the actual number of species for which we found occurrence data in GBIF, which we now do report (see first paragraph of “From species to ecoregions” in the Methods section). This latter number ranged from 2110 to 2714 species, representing 5-6 % of the woody plant flora.

Defining what is an acceptable threshold for the proportion of the flora that should be included is somewhat subjective. A way forward is perhaps comparing our dataset to other functional biogeography studies in the literature. In a recent global study, (Wright *et al.* 2017) evaluated 7,670 plant species, which represents less than 2% of the total global flora (i.e., 420,000 plant species; see (Govaerts 2001); see also (Pimm & Joppa 2015) for which this number is 450,000). In another recent study, (Bruehlheide *et al.* 2018) trait-environmental relationships at the global scale were addressed using 26,632 plant species (actually taxa; and this number refers to the total number of taxa across all traits, not for a particular trait, which was generally less). This represents 6% of the estimated global flora. Another recent global study with hydraulic traits and maximum tree height (Liu *et al.* 2019), using only data from woody species, included 1,281 species (3 % of the woody flora). Diefendorf *et al.* (2010) studied global patterns in leaf 13C content using only 334 woody plant species, from an estimated total of 175,500 (based on 39% of 420,000), totalizing much less than 0.5 % of the global woody flora. Therefore, the proportion of the flora represented in our dataset was within ranges generally accepted for this type of study and is fairly reasonable considering that Neotropical vegetation is much less studied and also more diversified compared to the global North.

Regarding our Leaf Spines database, which was based only on palm species, we recognize that the proportion is much smaller (694 species in total compiled; approx. 1%). However, it is the most phylogenetically controlled of all datasets and, despite of only including palms, there is a strong correlation with species richness for all woody species ($r = 0.75$; in relation to data from Kier *et al.* 2005). While it is true that palms are more diversified in forests, palms occur in almost all biomes in the Neotropics and are common in all Neotropical savanna ecoregions of South America. They usually dominate in savannas occurring in shallow and moist soils, which is the type of savanna dominating in many ecoregions, such as the Llanos savanna, the Guianan savanna, and Southern Cone Mesopotamian savannas of Argentina. Palm savannas are also common in savanna patches within the dry forest dominated Argentinean Chaco, especially in the Wet Chaco (you can visualize some photos here: <https://ecoregions2017.appspot.com/>).

In fact, palms and even palm-dominated savannas can be found including in the Cerrado ecoregion, the largest savanna ecoregion in the Neotropics. In this ecoregions, palm-dominated savannas are generally associated with shallow soils near water bodies and are named after the dominant palm species as “Vereda” or “Buritizal” (when dominated by *Mauritia flexuosa*, whose popular name is “Buriti”; the presence of a characteristic shrub layer near the ground differentiates the Vereda from the Buritizal),

“Macaubal” (dominated by *Acrocomia aculeata* or “Macaúba”), “Babaçual” (*Attalea speciosa* or “Babaçu”) and “Guerobal” (*Syagrus oleracea* or “Gueroba”). In addition, one of the largest compilations of the Brazilian Cerrado ecoregion flora (Mendonça *et al.* 2008) reports the occurrence of 59 palm species, of which at least 40 occur in some type of grassland, savanna and/or woodland vegetation. Many of those assume a morphology that is similar to the co-occurring woody species from other families, such as the dwarf palms *Syagrus petraea*. All the typical savanna palm species cited above are present in the Palm Trait Database used in this study.

While it is true that palm species richness is higher in forests, especially moist forest, we should not take a simple correlation as evidence of causation. Most likely, this pattern only reflects the well-known latitudinal gradient in species richness. In fact, it is not only palms that are more diversified in moist forest, but plant species in general, again, reflecting the latitudinal gradient in species richness (Kerkhoff *et al.* 2014). This interpretation is consistent with the correlation reported above for the richness of palms in our Leaf Spine dataset and that of woody species in general across Ecoregions in the Neotropics from Kier *et al.* ($r = 0.75$). Moreover, most of the variability in palm species richness is explained by precipitation and history, explaining between 71-82 % of the variability in palm richness (Bjorholm *et al.* 2005).

That being said, we are happy to inform that we made an extra effort to compile more data. We began by searching for data on leaf spines for other woody species. We found data for 2,173 woody species. However, a detailed inspection of this data showed that leaf spines is not a very common leaf defence trait outside of the palm family in the Neotropics. While we found 221 out of the 2,173 species to have spines, 198 of them were palms. Moreover, all but three non-palm species with leaf spines also had stem spines, indicating that, for other taxa, leaf spine is highly dependent on the presence of stem spines. A somewhat different situation was found for palms, for which only 101 out of 197 species with leaf spines (51%) also had stem spines. Indeed, there are numerous ways by which a woody species can protect its leaves against herbivores, and leaf spines is just one of them, and this trait is very plastic (Gödel *et al.* 2016). Many species rely on chemical defences, rather than spines (Coley & Barone 1996). Therefore, we concluded that using data for palm species alone was more informative of a megafauna effect in this specific case and therefore, we decided to maintain the use of the palm data, despite of having compiled data for other species. All arguments above suggest that palms, as a single clade having many species with spines, is a good candidate to track the processes we were interested in, that is, to use palms as a tracer of the footprint left by herbivores on plants.

In addition, we also decided to search for data on the presence of latex, which is not exclusive to leaves, but that is probably more useful against folivores. For this, we used the names of species already included in our trait dataset. We were able to find data for 3,160 species and have **now included this new trait** in our study. We also compiled more data on stem spines and were able to increase our stem spines dataset from 2,520 to 2,843 species. As you may see, the main patterns remained for this trait, although, now, a larger variance is explained by soil fertility and some effects associated to extant mammals are now evident. This change was also a result of a modification in the extant megafauna species index, which now excludes frugivorous,

granivorous, etc. (i.e., only include leaf and stem browsers and grazers). There was also a change in the relative importance of megafauna body mass compared to soil pH because of a bug in the package “dominanceanalysis” that was only identified during revisions. However, the values of the standardized coefficients in the previous submission already pointed towards a higher importance for pH compared to megafauna body mass. The problem with the “dominanceanalysis” package was that the functions `dominanceAnalyses()` and `averageContribution()` always used some parameter taken from the model that was run last, rather than the model that was specified in the arguments. Therefore, depending on the model that was run last, the function was giving different results; fortunately, this has now been fixed. Including latex in multivariate trait analyses also changed the correlation expressed in the main PCA axes, isolating the unexplained variance of leaf spines in a separated dimension that did not contribute to separate the antiherbiomes. However, the overall patterns were maintained for antiherbiomes, consistent with a figure presented in the previous submission indicating that leaf spines had little impact in these results (former Fig. S2).

As with Leaf Spines, the variability in Latex that was actually explained by our variables was low, although slightly higher, consistent with the idea that alternative mechanisms of leaf defence exist (Coley & Barone 1996) in such a way that no single trait is likely to capture all the variability that is driven by herbivores. We also show that latex was better explained by extant than extinct mammal herbivore distribution and tended to be associated with large-leaved plants from stable forest ecoregions.

We also searched for data on specific leaf area, even though previous evidence do not support this trait as an important defence trait against megafauna (Wigley *et al.* 2018; Dantas & Pausas 2020). However, we could only find data for approximately 1,800 species and, therefore, decided not to include it to avoid criticism regarding sample size. Yet, our analyses with this data confirmed our previous results indicating that past megafauna distribution did not influence current geographical variability in this trait. We also obtained data on maximum height for 20,000 woody species (Oliveira-filho 2017), but also did not find it to be influenced by megafauna. For simplicity, we decided not including these traits in the present study.

Regarding leaf size data, we did consider following your advice and look for data from herbaria, but we end up concluding that it would be very inefficient (high effort for little improvement of the dataset and results). In the best scenario, that is, one in which herbaria data was readily available from the internet (which does not seem to be the case for most countries in the region), obtaining leaf size data from an herbarium source would require making individual searches for each species, downloading photographs, running an image processing program (e.g., ImageJ), scaling it and calculating the area. We estimate that this would take at least 10 min per sample; or (being optimistic) about a month (full dedication) for processing ca. 1,000 samples that may be 1,000 species without replicates or much less if we consider the need for replication (for example, 250 species with four replicates per species).

Regarding wood density, we decided not to add extra data from the literature because all the extra data we found was for wood density measured in branches, whereas the data used here (from Zanne *et al.* 2009) was taken from references that

measured wood density in the main stem. Including data based on branches would introduce a lot of unexplained variability associated with these different methodologies. Moreover, the fact that we use a dataset that has been used in numerous previous studies on wood density prevents the interpretation that the differences in the results were explained by the inclusion of other species, rather than because of not considering all candidate hypotheses.

Finally, please notice that all this effort to make a synthesis involving multiple traits may preclude a very good representation of individual species for all traits, but can provide key insights in ecological processes that cannot be gained studying only one or a couple of traits.

Data for mammals: the data used for past megaherbivores relies on the co-occurrence with extant animals in the fossil record. However, there is another issue not presented, which is how present distributions of extant species (and thus past species) are inferred, called ‘range modifications’ in the Phylacine data. This should be acknowledged, as well as the limitations, and how it affects the conclusions (which modification is used, and why). In addition, the authors define megafauna as >50 kg – why? This does not follow the existing definitions (>40 or 44kg, or >1000 kg – especially when it comes to savannas the >1000kg may be a better proxy, similar to elephants), and it would be essential to test how this cut-off value affects the results. How do results change when choosing >1000 kg? Or >40 kg? Or >60 kg? Do they hold in a similar way? Also, the authors compare results of past megafauna, to the effect of current herbivores on plant traits (finding no significant effect, and thus concluding that present-day traits are largely the result of past interactions). However, for present-day herbivores, the authors do not focus only on megafauna. So datasets differ in two ways (extinct vs. present natural; megafauna only vs. all), instead of one. I suggest also focusing on megafauna only for this ‘sensitivity’ analysis, to make them comparable.

R: Regarding extant species and the “range modification” scenario, we added some extra clarifications in the Methods section (see “*Historical Megafauna Distribution*” and “*Herbivore Mammal Distribution*” in the Methods section). We preferred using this scenario because it is more likely to represent the herbivory pressure under which plant assemblages formed.

Regarding the threshold used to define megafauna, it is true that the use of 44-45 kg is common. Yet, many authors use other values to refer to megafauna, such as 5, 100, or 1,000 kg (Gill 2014; Ripple *et al.* 2015; Berzaghi *et al.* 2018). As highlighted by Galetti *et al.* (2017), these values are not used based on any biological criteria, but rather arbitrarily. Considering this, 50 kg sounded as a parsimonious value for us, as it is close to 44 or 45 kg used by many authors. To confirm that this decision did not affect our results, we calculated extinct megafauna species richness using the thresholds of 40, 44 and 60 kg and tested for correlations with our index (based on a 50 kg) across ecoregions. In all cases, the correlations were above 0.99 (most often 1.00, and in all cases, $P < 0.0001$). Therefore, this decision is unlikely to have affected the results in important ways. **These analyses are now reported in the Supplementary Methods**

(section “*Historical Megafauna Distribution*”). Also note that the decision of not choosing a very large value, such as 1,000 kg, is based on biological principles as it is well-known that mid-size mammal herbivores (generally considered to be mammal with approx. 250 kg) also have important effects in vegetation structure and function in Africa (see, e.g., (Charles-Dominique *et al.* 2016)), so it would not be appropriate for our question.

As for your last concern, which regards the use of extant mammal rather than extant megafauna (> 50 kg) species, you probably know that most megafauna species from South America are now extinct, and none of the extant species are browsers. In our dataset, there were only 13 extant mammal herbivore species with more than 50 kg in the Neotropical region, from which nine are grazers and three are mixed-feeders. Thus, such an index is not likely to provide useful information in this context. Also remember that extant mammal body mass was also a candidate variable in all the models, allowing the detection of potentially important effects associated to animal size. In fact, our new results show that this variable is a significant driver of latex production, with this trait being more favoured where extant species are small (Fig. 2).

Averaging for bioregions: first, a bit more detail on the bioregion classification needs to be provided, as this is the unit of analysis, and there is a lot of different opinions about this classification. How sensitive are results to this classification? Second, why use the ca. 60 (or so?) bioregions as unit in the analysis, why not use the grid cells the data comes from directly? This would increase the sample size, avoids making arbitrary averages, and the interpretation would not rely on ‘bioregion’. I also wonder whether a cut-off value was used to include the unit in the analysis, for example, it should have data for at least 100 plant species occurring in there? Or at least 10? Because if it’s based on 1 species, it’s clearly biased...

R: We are not sure where this value (ca. 60), mentioned by the reviewer, came from. Most of the trait analyses were run with 142 (131-157) ecoregions, from a total of 179 ecoregions in the Neotropics (see Table S2 in the previous version). The number of Ecoregions included in the trait analyses was limited by 1) availability of megafauna and environmental data for the ecoregion (available for only 157 ecoregions); 2) by the exclusion of islands (justified in the methods); and 3) for leaf spines, by the distribution of palms. Despite of that, we had plenty degrees of freedom do address all our hypotheses. Also note that we corrected one missing value for extant mammal species in our data matrix and now have one more ecoregion included in the trait analyses (see Supplementary Table 2 of the current version).

We have now provided more details on the ecoregion classification scheme, providing a clear definition of Ecoregion in the end of the introduction: “Ecoregions are regions characterized by distinct natural assemblages of plant species whose boundaries are defined based on detailed vegetation and flora surveys, as well as expert opinion²³”.

Note that given the definition above and the scale of our study, the ecoregions is probably the best available approach. Unlike other approaches, like grid cells or botanical countries (Lim *et al.* 2020; Onstein *et al.* 2020), our approach justifies conceptually. One cannot simply take a different result between these different

approaches as evidence that one is better. It is true that using grid cells we would increase our sample size (in fact we can increase it as much as we would like by using small cells and downscaling), but would also largely increase spatial autocorrelation and type I error (false positives). Ecoregions are defined to be more homogeneous within than between and thus they drastically reduce spatial autocorrelation. Moreover, using grid cell can be misleading, because it could lead to misrepresentation of the dominant vegetation type. In the Neotropics, there are many enclaves of other biomes occurring as mosaics side by side with the dominant type and biases related to this specific pattern should be more common at finer scales.

In addition, the idea suggested by the reviewer that results based on few species are necessarily biased is not straightforward. Some of these ecoregions are deserts, which contain very few woody species, others are naturally characterized by the dominance of few woody species, and others are distant small islands, for which island biogeography theory predict low species richness. On the other extreme, we have highly biodiverse forests. For example, for wood density, we found five ecoregions to be represented by 10 species or less. From these, three are deserts or semi-deserts (Atacama Desert, Southern Andean steppe, Santa Marta Paramo), one is strongly dominated by pine and oak species (pine-oak forest), and one is in a distant small group of islands (Fernando de Noronha). A similar pattern was observed for leaf size, with 3 ecoregions with 10 or less species, one of which is a desert (Atacama) and the other two which were distant islands (Fernando de Noronha and Juan Fernández). Moreover, regions that are naturally species-poor because of strong environmental filters (e.g., deserts, mountain tops) are expected to present lower trait variability, in such a way that fewer species are required to represent the overall trait values of the flora.

For leaf spines, for which we used only palms, the number of ecoregions in this category was higher, but since it is a complete palm species database, the low species number is probably also a natural pattern, rather than an artifact. Thus, the few occurring species within the ecoregion can be considered as representative of how the palm group manifests under those conditions (but as you may see, this trait had little influence in the overall patterns of the antiherbiomes). Thus, we do not believe that adding an analysis that excludes ecoregions with fewer species would provide any additional information. Especially because, as previously shown, there is a strong correlation between species richness in our dataset with that previously reported by Kier et al. (2005; see previous replies in this Letter).

I would be interested to see the raw data for these analyses, the plant-species matrix by traits, as well as the species x bioregion dataset for both plants and mammals. Please submit this if invited for a resubmission/revision.

R: We have now added this data to the submission (Supplementary Tables 11-23)

Data analysis: the study uses simple linear and generalized linear models to associate plant trait averages in bioregions, to mammal richness, climate and disturbance (fire

etc.) variables. It uses step-wise model selection, and then assesses the relative importance of the variables. It is an ok approach, but a much more elegant approach would be to use structural equation models, especially because the authors talk about potential direct and indirect effects, which could be assessed in these models (as well as different response variables, so all traits could be evaluated in a single analysis). Regardless on whether SEMs are used, it would be important to have a distance measure in the model to correct for potential spatial autocorrelation (or evaluate with a SAR model).

R: The use of models that account for spatial autocorrelation is only appropriate if one detects spatial autocorrelation in the residuals of the analyses. We have now tested the residuals of our selected models for spatial autocorrelation using the Moran's I statistics. None of the models showed significant spatial autocorrelation in the residuals (as was expected when using ecoregions; see above) indicating that no further analyses using spatial autoregressive models were necessary. These results are now show in Table S4 and described in Methods (see "Statistical Analyses").

We appreciate the fact that the reviewer understands that "elegance" is subjective. We agree that elegance is important, but the key point is whether our analysis is wrong, and it doesn't seem to be the case. In fact, reviewer 2 found the statistical approach to be "current and well designed". While we recognize that SEMs can be very useful in some contexts, they can also be very complicated when involving several variables and not everyone is familiar with it. We also preferred to maintain methods similar to those that have been used in recent related studies published by this same journal (Lim *et al.* 2020).

Data analysis – sensitivity analyses/simulations: Second, as mentioned, the study is correlative, and correlations could arise because of potential other (unmeasured) factors. It would therefore be essential to do several sensitivity analyses. For example, when randomising trait averages (the response) across bioregions 1000x times, does the observed standardized effect come out as unique (outside 95% probability of simulated datasets?) And when using a trait as response that is not supposed to relate to past megafauna, is megafauna indeed not significant? For examples, see here: https://royalsocietypublishing.org/doi/full/10.1098/rspb.2019.2731?url_ver=Z39.88-2003&rfr_id=ori:rid:crossref.org&rfr_dat=cr_pub=pubmed

R: We have now included the analysis suggested by the reviewer based on 1,000 randomizations of the ecoregion matrix for all herbivory related variables that were selected in a given trait model. None of the herbivory related variables that were significant in the original regression model was not significant with the approach suggested by the reviewer. These results are now presented in Table S5 and the details are described in the Results and Supplementary Methods (in the subsection "Statistical analyses") sections.

Also, we only compiled data for traits that we initially thought that could be related to megafauna. As previously reported, we have now compiled data on some other traits for

which there is not clear evidence of being substantially influenced by megafauna (for example, maximum tree height and specific leaf area) and those were, indeed, not found to be related to herbivory. However, some of these results were inconclusive (SLA) and for simplicity, we preferred not including these traits in the study. Yet, if the reviewer present convincing arguments that this was strictly necessary, we may reconsider. However note that what consists of a trait that “is not supposed to relate to past megafauna” is not a very objective matter; tradition also plays an important role (Pausas & Bond 2019). Decades ago, many ecologists would be confident that spinescence had nothing to do with herbivory and was, rather, an adaptation to reduce water loss in arid ecosystems.

The discussion feels a bit ‘light’ and could do with more thoroughly relating results and interpretation to the existing literature, and especially make a more in depth comparison to Africa (and Asia?).

R: We have rewritten most of the discussion and made an extra effort to better relate the patterns observed with previous results from Africa, when available. The effects of herbivores in Asia are much less understood.

Minor comments:

Use more informative headings in Supplementary information tables and figures, explain for figures what can be seen and some sort of interpretation, and explain ALL abbreviations in tables. Also in the main text some abbreviations in figures are not explained in caption. Abbreviations are generally quite confusing, and it would be more helpful to avoid them where possible.

R: We have revised all legends in the manuscript as well as in the Supplementary Information, adding more detailed information wherever needed. We also made an extra effort to check if all abbreviations were explained in Figures and Tables, and avoided abbreviations in the text as much as possible.

Table S3 MAR \diamond MAP

R: fixed. MAR is the form we are preferring.

Why are abiotic variables often not significantly explaining defence traits in the linear and generalized linear models, but then are included when defining the ‘antiherbiomes’ based on cluster analyses? Are they suddenly correlated to them in the cluster analysis, and why? Perhaps explain better.

R: No, they are not. These associations were rather used to allow a clear comparison with patterns for Africa. To make it clearer what are the variables that actually explain these patterns, we added regression analyses on the principal component axes that

separated the antiherbiomes (Fig. 5; Table S3). We also added to the discussion section the following text:

“Consistent with this idea, regression analyses for the Physical-Chemical and Thorny-Woody axes suggested that only nutrient availability and megafauna history explain these large-scale patterns (Fig. 5). However, it is important to bear in mind that, in our dataset, soil pH and mean annual precipitation were negatively correlated ($r = -0.78$; Supplementary Table 1) and, thus, these two variables were not entered in the same model to avoid multicollinearity (see Methods).”

“There are multiple mechanisms by which plants defend themselves from large herbivores 5,7,11–13, and these mechanisms differ with climate and availability of soil resources” – please describe in more detail how they interact.

R: We actually meant “traits” here, rather than “mechanisms”, so we corrected the sentence to:

“There are multiple traits by which plants defend themselves from large herbivores 5,7,11–13, and the dominant traits often differ with climate and availability of soil resources 12–18”

The details are provided in the following sentences but note that the actual mechanism relating traits and the environment in Africa are not fully elucidated. Nevertheless we have now added one important hypothesis (Bell 1982), which is consistent with our results:

“While the causes of these trait-environment associations had not been rigorously addressed, evidence points towards climate and soil as key factors mediating herbivore activity via changes in plant tissue nutritional quality²⁰.”

“mesic nutrient-poor savannas dominated by broad-leaved plants that mainly rely on leaf defences” – normally, low nutrients lead to small leaves and low specific leaf areas (sclerophylly) – this is not the case here? Why not? This contrasts with several other low-nutrient biomes and could be briefly discussed.

R: In the tropical context, a small leaf is a trait related to xerophily, a syndrome of arid nutrient-rich environments. This is true for Africa, for which it is a very well-known pattern (Bell 1982; Wigley *et al.* 2018). This can also be readily depicted for South and Central America, from a comparison between our current Fig. 1g and Supplementary Fig. 3a,c. In these maps, you will see that smaller leaves are associated to nutrient rich soils, which is what we expected based on own experiences.

Moreover, whereas low specific leaf area (SLA) can be sometimes associated with sclerophylly, this trait was not formally addressed in this study, as there is little data available and also little evidence that it is effective against megafauna (Wigley *et al.* 2018; Dantas & Pausas 2020). This trait responds to a multitude of drivers, including

nutrient, water and light availability, and have even been suggested to be under selection by fire (Pausas *et al.* 2017; Dantas & Pausas 2020), and therefore, we do not consider it to be a reliable trait to study these patterns. This is not the case of Leaf Size, which was the trait that we used.

“A third ecosystem type can be readily identified in tropical Africa, that is, forests, in which low levels of megafauna herbivory and high productivity enable quick canopy escape, and, therefore, plants are largely undefended” – hm, not sure if I believe this, many rainforest plants have hooks and spines, often also related to climbing. Especially in palms (one of the clades focused on here...)

R: It is a bit risky to infer a process from those simple observations; note that our results suggest that many of the spines in forests may come from Pleistocene savannas. In any case, we have now changed this prediction to better account for previous studies showing that chemical defences are important in forests, a pattern that is consistent with our results for latex.

“large mammal herbivores over 50 Kg that became extinct in pre-historical times” – define what pre-historic times are.

R: We have now clarified this in several points and in the Methods (see “Historical Megafauna Distribution”). In this dataset, the period refers to extinctions that occurred prior to 1500 CE up to 130,000 years, matching the IUCN criteria.

“Woody density was especially high and leaf size especially low where extinct large grazer (i.e. megagrazers) richness was greater than that of large browsers (i.e. megabrowsers; Fig. 2; Table S3, S5).” – at this point in the text it is not clear why distinguishing grazers and browsers, clarify here or before.

R: We have now clarified: “The occurrence of more extinct megafauna grazers relative to browsers was an important driver of investments in defence such as wood density and spines (Fig. 2; Supplementary Tables 3 and 4), highlighting the important role of typical savanna megafauna”. The details are provided in the third paragraph of the discussion.

“In contrast, extant mammal herbivores did not explain any proportion of the variability observed in antiherbivory defence traits (Fig. 2; Table S5).” – I do not see this in Table S5 (This seems to only relate to using megafauna as response? It would be good to have the comparison.) – not clear which tables relate to which analyses.

R: These results have changed because we have modified the extant mammal richness and body mass indices to exclude animals that did not feed on shoots (e.g. frugivorous; to address comments by the second reviewer) and expanded our spinescence database, which resulted in a large increase in the variability explained in spinescence (R^2_{adj} increased from 0.38 to 0.53). Also note that, as described in the methods section, both

megafauna and extant mammal predictors were included in the initial full models. Those variables that do not appear in the final model should be interpreted as variables that did not show a significant effect.

“the relationship with extant herbivore richness indicated bottom up control of these traits on modern mammal distribution (i.e. a significant negative relationship; Table S7).” – hard to follow, please explain.

R: Removed from current version.

“A cluster analysis on the trait by ecoregion matrix (see Methods for details) supported the hypothesis that three distinct antiherbiomes can be recognized in the Neotropics.” – bit more details on the analysis needed (without having to read materials/methods first)

R: We tried to make it clearer, see third paragraph in the Results section.

“(3) megafauna sensitive assemblages” – confusing, what does ‘sensitive’ mean or refer to?

R: We no longer use that term.

“are not under strong selection during the Holocene”- Why not? They are still there, and correlate with past megafauna?

R: We have removed this sentence from the text.

“where grazers were relatively more abundant (Fig. 2)” - Compared to browsers, make clear

R: This was modified, but we made sure to provide this information when appropriate paragraph (see third paragraph in the discussion).

“suggesting that this defence trait is especially adaptive under grazer-controlled fire regimes.” – why fire?

R: We have removed this sentence from the text.

“In Africa, large browsers and mid-size social mixed-feeder species are important predictors of stem spines, although the latter are presumably more important 12. In our study, including mixed-feeder did not improve model fit (not shown).” – please do show

R: We have now reported the AIC differences of models with and without the richness of mixed-feeder species (see last phrase in the ninth paragraph of the discussion)

“that leaf size is especially useful as a defence mechanism in nutrient-rich soils and drier climates.” – please repeat why

R: This sentence is no longer in the manuscript. While the mechanism was yet not fully understood (only the pattern was well-known), we now discuss with more detail potential mechanisms throughout the discussion, especially in the second paragraph.

“For many species we had more than one trait value, so we computed the species trait value as the mean for quantitative traits” – for how many?

R: We have now added this information to the Methods section (see last paragraph of “Plant defence traits”). For wood density and leaf size, there was 1005 and 831 species, respectively, with more than one trait value. For binary traits, in which mostly herbaria data or published datasets were used, this information is often not provided. For instance, for Leaf Spines, we directly used data from the palm trait database which does not provide such details. This was also the case of stem spines, in which it was often described in herbaria that a species sometimes present spines (suggesting that the information is based in more than one individual), but the exact number of individuals evaluated was unknown. Since the information could not be retrieved in all cases, if data from a data source reported a value for a species that we already had information in the dataset and this was a different value, we directly replaced the value in the compilation table without recording in which species this occurred and how many times. However, this happened in very few occasions, as most of the binary data came from one single source containing only general information for the species, rather than for plant individuals. Moreover, we searched for data using the name of species already in the dataset (with data for other traits) and for which we did not already have information on the targeted trait.

“We obtained geographical distribution data (coordinates) from GBIF for all of the species in each species-trait dataset” – Cleaned? How dealaed with biases? Taxonomical issues?

R: Taxonomic issues were dealt with during the process of making data queries to the GBIF database. In the R codes used for this, which were directly suggested by the gbif helpdesk, the queried names were matched to the GBIF backbone, which contains former classification names for every taxon. The search returns both the original search and the corrected names, only returning data for which the names were accepted or matched to the backbone. The code also removes anything that might have matched to a non-plant. Subsequent analyses were carried using the suggested species names. This process resulted in a reduction in the number of species included the study and we now report the number of species that was effectively considered (see “From Species to Ecoregions” in the Methods section).

“Because palms were missing from 20 ecoregions, we completed the values for these ecoregions using predicted model probabilities.” Bit tricky when based on palms only...

R: please, see comments about palms in our second reply to you.

Renske Onstein

Reviewer #2 (Remarks to the Author):

The paper proposes an interesting analysis of the effect of large mammal herbivores on the evolution of plant traits and their biogeography in Central and South America. The authors show the presence of 3 distinct regions or “antiherbiomes” which are characterized by similar plant defense strategies, megafauna history, and environmental factors. They also identify regions which have shifted from being grass-dominated to forest-dominated. Their results suggest that herbivorous megafauna had a significant and long-lasting effect on plant traits and biome distribution.

The topic is both interesting and relevant to the increasing focus on the study of the ecosystemic role of large terrestrial mammals. In this context, the research questions addressed by the authors are of great importance not only for plant-animal ecology but also for other related fields such as evolution and biogeographic modeling. The methodology makes good use of a lot of different data sets and datatypes by including trait data of both animals and plants. However, I think there are some key points to be addressed or clarified to improve the paper. The statistical approach is current and well designed. The figures are nice and the results well presented also in the supplementary. The discussion section is also interesting and well written but I think there are a few important points that are overlooked.

R: We thank you for your encouraging and supportive comments and hope that the revisions have addressed the issues raised.

One of my main concerns is that the authors relate the effect of megafauna to plant traits based on correlations between species richness and mean body mass. However, herbivore population density is probably just as important than species richness or mean body mass because population density determines the intensity of “animal disturbance” on the landscape and on plants. For example, during the Pleistocene in the tundra-steppe ecosystem the species richness was low but the density of mammoths and other large herbivores was probably quite high and thus it had a significant effect on the ecosystem. In tropical grasslands the herbivore biomass should be higher than tropical forests. Similarly, for mean body mass, a higher mean body mass doesn’t necessarily relate to a higher animal biomass per area. The relationship between body mass and density can vary considerably. Considering that herbivore density is a key parameter in plant-animal

interactions I wonder how the absence of such factor might influence the authors' results.

R: Unfortunately, there aren't good methods to estimate the density of extinct megafauna species. The main methods currently rely on allometric equation based on body mass to estimate population density (i.e. thermodynamically based trophic models) or secondary productivity (i.e., productivity models; see (Giacomo & Fariña 2017)). However, using these equations for our Neotropical data resulted in indices that were strongly positively correlated to megafauna species richness (Supplementary Fig. 3). Since this metric was also derived from body mass, we preferred using richness and body mass directly, as these variables are also likely to capture additional dimensions of herbivore effects. These analyses and indices, as well as the motivation to consider them, are all now described in the Supplementary Methods (see "Historical Megafauna Distribution").

Whereas the relationships reported in Supplementary Fig. 3 generally supports our assumption that megafauna richness is probably a good proxy for megafauna density and productivity, animal density is also likely to be influenced by soil fertility (McNaughton 1988) and, therefore, these metrics likely captured only a fraction of the variability in megafauna density and productivity. However, note that we also included soil variables and our interpretations of the soil results always take their effects on animal density/productivity into account. In fact, as now extensively argued in the Discussion (see, e.g., the second paragraph), this is likely to be the mechanism underlying the relationships involving soil pH and the defence traits.

Therefore, while we did not include accurate density estimates explicitly, we argue that, combined, our variables are able to capture this effect. Moreover, even assuming that none of our megafauna variables really captured these effects, including accurate density estimates would be more likely to increase the fraction of the variability that was explained by megafauna, rather than substantially change the conclusions. In any case, please consider that it is not as easy as it is to extant mammals' herbivores to produce a reliable map of extinct megafauna density and such maps are not yet available for the Neotropics, as far as we are aware of. We expect that our study stimulates the development of this type of dataset allowing for a deeper investigation of these ideas.

Another unclear point is that even though herbivores have been divided in grazers, browsers, and mixed-feeders, there can be substantial differences intra- and inter-species depending on the ecosystem. Extinct megafauna in the Neotropics probably consumed large amounts of fruit in forested areas. In those cases, plants probably invested in other key traits or defenses. In this regard, the authors should clarify their definition of herbivores because in the Phylacine database herbivores are not differentiated according to their plant dietary preferences (fruits, seeds, leaves). I do not see how including all these different herbivore guilds is relevant to the plant traits used in the analysis.

R: Indeed, the Phylacine database only brings information on how much of an animal's diet consists of plants. Our first cut was excluding species for which plants was not at least 90 % of the diet. In the next step, we selected species with at least 50 kg (for extinct megafauna, but not for extant herbivores) and calculated the richness and body mass indices, as well as the per diet richness, for the species for which we found data. While for extinct megafauna species detailed guild information is not available, for extant mammal species, in contrast, the diet dataset used explicitly differentiated between frugivorous and animals feeding on leaf/branches (grazers and browsers, in this dataset). Therefore, for extant mammals, we have now recalculated the richness and body mass metrics considering only the species for which the diet was primarily based on leaves and stems (details are now provided in "Herbivore Mammal Distribution" in the Methods section). This changed some of the results and extant mammal herbivores now explain some of the variability in spinescence, as well as latex, a new trait included.

For the extinct species, grazers and browsers are differentiated based on isotopes and/or morphology (see details added in "Historical Megafauna Distribution" in the Supplementary Methods), and current methods do not allow to identify predominant frugivorous (Lim *et al.* 2020). Therefore, we assumed that none of the extinct megafauna browsers were frugivorous, an assumption that is fairly reasonable, if you consider that animals with 50 kg or more are rarely frugivorous in Africa (see Hempson *et al.* (2015)). Also note that, like in most studies of the type, grazers and browser are classified according to the dominant diet and are called mixed-feeders when a predominant food source cannot be identified. We also classified as mixed-feeder those species that were found to be browsers in one site and grazers in another site.

A recent study produced a tentative list of frugivorous extinct megafauna species (Lim *et al.* 2020). However, their definition of frugivory was not based on dietary proxies, but rather in the assumption that extinct and extant species from the same family mostly shared similar diets. Thus, if more than 50 % of the extant species in a taxonomic family fed predominantly on fruits, the extinct species was also assumed to have that diet. This is problematic for megafauna species, as they are rarely frugivorous and there is a lot of intrafamily variability (Hempson *et al.* 2015). For instance, they classify elephant-like species of the family Elephantidae as frugivorous, but, in Africa, this family includes the extant keystone species *Loxodonta africana*, the savanna elephant, a mixed-feeder that feeds on leaves and stems (Hempson *et al.* 2015). Accordingly, this list includes species that were found in our compilations to be predominantly grazer, such as *Notiomastodon platensis* and *Mammuthus columbi*.

Therefore, while we now provide more details on diet discrimination (see "Historical Megafauna Distribution" in the Supplementary Methods) and recognize these limitations in the discussion (see end of the paragraph that precedes the last one in the discussion), we believe that this issue is likely to have little effects in our results.

I don't completely agree with the authors discussion in regards to leaf size and wood density. Wood density is related to many eco-physiological properties and has high

intra-specific variation. Authors conclude that wood density is higher where grazers were relatively more abundant but the wood density patterns presented in Chave et al. 2009 do not confirm this trend. There is also a trade-off between growth rate and wood density. Lower wood density is associated with increased growth rates and potentially provides a faster escape from the herbivore trap at the cost of reduced mechanical strength. In regards to leaf size, Wright et al. 2017 state that differences in leaf to air temperature are key to shaping leaf size, I think this should also be discussed. Also, the Wright et al. included a large range of growth forms, so direct comparisons might be tricky. In general, I find it somewhat confusing that only woody species are discussed because grazers and mixed-feeders are associated with feeding on herbaceous vegetation and thus the antiherbiomes are only described in terms of woody species traits but grasses also develop herbivore defenses.

R: Chave *et al.* (2009) is an article on trait-trait correlation and it is often difficult to infer causality based on correlations. Correlation between wood density and other physiological traits can both indicate physiology as a driver or woody density as a constrain on plant physiology.

It is not clear to us how the reviewer concludes that the results presented by Chave et al. (2009) are not consistent with the presence of grazers as a factor associated to high wood density. Perhaps the author is here referring to the map of wood density shown in the aforementioned article (Fig. 6 in Chave et al. 2009), meaning that it does not fully match our grazer map. If so, first of all, it is important here to say that the dataset used here is the exact same one as dataset used by Chave et al. (2009), that is, the Zanne et al. (2009) dataset. We did not complement this dataset because we only could find additional data that measured WD in branches, rather than stems (which is the case in the dataset used by Chave et al. 2009). That being said, note that: 1) this effect of grazers to which you refer to is very small compared to other variables, such as megafauna richness, soil sand content and hurricanes, and is likely to be indicating a more important role for savanna herbivores (C4 grasses are shade intolerant), as well as an indirect effect, mediated by the favourable condition for the development of these traits associated with ecosystems in which grazing occur; 2) In their Fig. 6, it is clear that very high WD is found in the Chaco region in Argentina and Paraguay, and the Chaco ecoregions were those for which grazer richness was highest in our map (Fig. 1c).

Regarding the evidence shown by Chave et al. (2009) that growth rate decreases with wood density, please note that this specific relationship was derived from saplings with 1–5 cm of stem diameter in two forest sites. Moreover, many studies are showing that relationships between traits that were previously shown for forest species do not necessarily apply to other ecosystems. For instance, Wright *et al.* (2018) found no evidence that growth rate and wood density are related in Australian savannas. Moreover, many of the previously reported relationships between leaf traits at the global scale are also not that evident in Afrotropical savannas (Wigley *et al.* 2016). Just because a relationship applies in one ecosystem or one scale, it does not mean it applies to other ecosystems and scales. In the context of forest, much of the variability in wood

density may be explained by the different successional positions of species, with pioneer species growing faster than old growth species (Berzaghi *et al.* 2019), but forest probably do not comprise the entire spectrum. Moreover, in the same way that we must be careful when extrapolating results that are based in tropical forest to other biomes, it is tricky to extrapolate something that is based in 1–5 cm sapling to adult trees.

We also believe that it is unlikely that very low wood density species can support very tall trunks. In a recent study, we have shown that Afrotropical savanna species have higher wood density and are taller compared to Neotropical savanna species (Dantas & Pausas 2020). Plants in the former savannas also seems to growth faster in height than the later (Dantas & Pausas 2013).

Regarding leaf size, Wright *et al.* (2017) only included climatic variables, which precludes a detailed comparison with our results. Many ecological patterns are correlated with climate in such a way that it is impossible to fully address the relative importance of climate without explicitly consider other candidate variables, such as soil and natural disturbance regime. Our study is pioneering in this aspect. Yet, notice that we address a different scale here and thus, the fact that other variables are more important in the context of the Neotropics does not necessarily imply that the same variables are important at the global scale. These aspects could have been better recognized for the associations shown here, so we now add a bit more discussion on this and also on the fact that we did not considered herbaceous species (see eighth and tenth paragraphs in the discussion).

We also provided an explanation in the Methods section for why we did not include leaf size for herbaceous species (see “Plant Defence Traits”). Since the mechanism by which megaherbivores influence leaf size in woody and herbaceous species are likely to be different, it would be more appropriate to analyse the data for woody and herbaceous species separately. However, there is much less data on plant traits for herbaceous compared with woody species in the Wright *et al.* (2017) dataset for South and Central America, including data for only 253 herbaceous species. This is less than two species per ecoregion and, therefore, we opted for not using it.

I think there should also be a paragraph to discuss the interpretation of these results in term of short-term and long-term adaptations. As mentioned by the authors in regards to spines, some of the plant traits included in the analysis can be very plastic and the lack of phylogeny correction in the analysis does not allow to fully evaluate the results in the evolutionary context. For a better evaluation of the evolutionary effect phylogeny correction should be included in the analysis.

R: We have added the suggested paragraph discussing how plasticity could have affected our results (see ninth paragraph of the discussion). Indeed, the focus here is showing the effects of megafauna in plant distribution, not evolution.

A few minor comments:

- From what I understand the term “Neotropical” is used throughout the paper when referring to South and Central Americas. I find this confusing because the terms

Neotropics and Afrotropics are often used in tropical ecological literature when referring strictly to tropical biomes. I'm not sure why the term Neotropical is used since this paper covers South America which includes a wide range of biomes.

R: The term Neotropical was used as a reference to the Neotropical biogeographic realm, originally suggested by Alfred Russel Wallace, but still largely used in the biogeographic literature for both plants and animals. Despite of the name, this realm comprises the entire South and Central American continents, including subtropical and temperate regions. The exact map that was used in this study to delimit the Neotropical realm can be found in <https://ecoregions.appspot.com/>. We used this map because it is based on biogeographic studies, rather on arbitrary political boundaries. We were divided about keep using the term as we also would like to use terms that are more familiar to the general reader. However, the term Neotropics is the one that most accurately match the region considered. In any case, when possible (e.g., starting from the Abstract), we have made clear that we refer to South and Central America.

- See annotated comments in text
- Diet studies of extinct herbivores often relate C3 plants to a closed-habitat browser diet and C4 to a grazer diet, however there can also be a mix of C3/C4 plants in the diet making difficult to assess. I would add a bit more information in your methodology on how you determined grazing and browsing for extinct animals.

R: We have now added more information in the Supplementary Methods, in the subsection "Historical Megafauna Distribution".

- The legend of figure 5 should be made more clear

R: We have now completely reformulated the legend. Also, this is now our Fig. 6.

- Please discuss more why the Amazon basin, the cerrado and the Atlantic Forest are under LDW and the Western Amazon is not LDW.

R: We have added the following statement to the discussion section, in the part that discuss the now called BCL antiherbiome (formerly: BLS): "This stability could be related to the high climatic stability of the region associated with orographic Andes-related rainfall, as well as the nutrient-rich sediments (high CEC) derived from the Andes Mountains, responsible for the clear water nutrient-rich rivers of the Amazon (as opposed to nutrient-poor dark water rivers)."

Some useful references

1. Berzaghi F, Verbeeck H, Nielsen MR, Doughty CE, Bretagnolle F, Marchetti M, et al. Assessing the role of megafauna in tropical forest ecosystems and biogeochemical cycles – the potential of vegetation models. *Ecography*. 2018;41(0):1–21.
2. Chave J, Coomes D, Jansen S, Lewis SL, Swenson NG, Zanne AE. Towards a

- worldwide wood economics spectrum. *Ecology Letters*. 2009 Apr 1;12(4):351–66.
3. Hatton IA, McCann KS, Fryxell JM, Davies TJ, Smerlak M, Sinclair ARE, et al. The predator-prey power law: Biomass scaling across terrestrial and aquatic biomes. *Science*. 2015 Sep 4;349(6252):aac6284.
 4. Santini L, Isaac NJB, Maiorano L, Ficetola GF, Huijbregts MAJ, Carbone C, et al. Global drivers of population density in terrestrial vertebrates. *Global Ecology and Biogeography*. 2018;27(8):968–79.
 5. Vizcaíno. (PDF) Evaluating Habitats and Feeding Habits Through Ecomorphological Features in Glyptodonts (Mammalia, Xenarthra). ResearchGate [Internet]. 2011 [cited 2018 Dec 5]; Available from: https://www.researchgate.net/publication/257931077_Evaluating_Habitats_and_Feeding_Habits_Through_Ecomorphological_Features_in_Glyptodonts_Mammalia_Xenarthra
 6. Zhu D, Ciais P, Chang J, Krinner G, Peng S, Viovy N, et al. The large mean body size of mammalian herbivores explains the productivity paradox during the Last Glacial Maximum. *Nature Ecology & Evolution*. 2018 Apr;2(4):640–9.
 7. Doughty CE, Wolf A, Morueta-Holme N, Jørgensen PM, Sandel B, Violle C, et al. Megafauna extinction, tree species range reduction, and carbon storage in Amazonian forests. *Ecography*. 2016 Feb 1;39(2):194–203.

Reviewer #3 (Remarks to the Author):

This paper presents a new analysis looking for evidence that the geographical some functional traits (plant defence traits) of South American flora can be largely explained by the presence of past megafauna. The approach taken is to combine (i) information on the spatial distribution of plant species (ii) data on species defence traits (spinescence, leaf size and wood density) and (iii) information on past distribution of megafauna (previously published by other authors) that infers distribution based on coexistence with extant fauna in the fossil record, and then using yeh current distribution of extant fauna.

The emerging insights are that (i) past megafauna can explain a substantial fraction of modern plant defence trait distribution(i) it is possible to identify three broad “antiherbiomes” in South America” based on plant traits, analogous to ones identified in Africa where megafauna are present. They also suggest there are regions that were grassy under megafauna presence but shifted to forest domination following megafauna extinction.

The novelty of this work is a clear quantification of the proposition that the extant flora of the Americas carry the “ghost” signatures of recently extinct megafauna, which enables both an interpretation of why these traits are found where they are, and opens up an avenue to refine understanding of the past ecology and distribution of the megafauna, and how South America’s ecosystems changes following megafaunal extinction. These topics have been the subject of much speculation and this paper would make a substantive analytical advance in the field. The paper would also influence our

understanding of South American ecology, which tends to ignore the possible role of past megafauna in shaping modern day ecosystems.

R: Thank you for seeing and summarizing the essence of our study.

Overall, I am intrigued by this paper and its argument, but feel it relies rather heavily on broad spatial statistical analysis, without critical interrogation of the datasets involved or the sweeping conclusions buried in the statistics and tables. Figures 1 and 2 illustrate this. A key part of the argument is here and warrants greater exploration. The most striking result is that most of the biomes of Brazil carry the signatures of the LDW antiherbivore (Fig 2). Looking more closely, this seems to be the conclusion derived from high levels of leaf spines in palms in this region, and higher wood density *Fig. 1). The prevailing ecological interpretation has previously been that high wood density is linked to soil properties in the crystalline shield region (deep soils, low fertility) compared to high fertility and shallow soils in the Amazon lowlands (see papers by Beto Quesada et al for Amazonia). I wonder if the soil variables picked here really capture this gradient, whether past megafauna richness really is the predominant driver of patterns in wood density across the Neotropics? I may be wrong, but I feel this requires more sceptical exploration, beyond simple reporting of broad spatially correlated statistics. How much do we trust the inferences of the PHYACINE database? Have the soil environmental variables been accurately captured? (no reference to the extensive literature on this)?

R: We have now incorporated the article of Quesada *et al.* (2012) in our discussion and added a detailed comparison with their results (see eighth paragraph in the Discussion). It is very interesting to notice that, while the study by Quesada et al. (2012) is based on stand level data and on forest, which implies reduced error compared to our approach, their results is broadly consistent to ours in what concerns the relationships of wood density, soil and climatic variables. Yet, while their study did not incorporate historical herbivory, preventing an accurate comparison of different hypotheses, in our study (that did), megafauna richness appear as more important. It is also worth noticing that the negative association between cation availability and wood density reported in this study is broadly consistent with evidence presented in our results that megafauna richness and cation exchange capacity are negatively correlated. We do not think, however, that this could be explained by wrong inferences in the PHYACINE database as this would be more likely to introduce error, rather than increase the variability explained by megafauna richness. Yet, we have added several comments on the potential limitations of the Phylacine in the tenth paragraph of the discussion.

Is Figure 2 really saying that much of Amazonia was a herbivore accessible mosaic in the Pleistocene? Or is it that sub-canopy farms had some herbivory pressure in what was still extensive closed-canopy forest?

R: The high abundance of grazers in eastern Amazon (> than the 75% quantile of megafauna grazer richness) is more consistent with this area being a savanna, because the

C4 grasses that dominate in the tropics are generally shade intolerant. The fact that these ecosystems were classified in the now called ILW antiherbivore (formerly LDW) which, as argued, closely resemble African mesic nutrient-poor savannas, is also consistent with this assumption. The fact that these areas were megafauna-rich, a pattern that is not usually associated with closed-canopy forest, also support these ideas. We have now added more detailed discussion on both, how the ILW antiherbivores match African mesic nutrient-poor savannas, as well as specific discussion about the Amazon (see fourth, fifth and sixth paragraphs of the discussion).

Lines 52-56: It would be helpful to provide some mechanistic speculation/explanation on why these different environmental conditions lead to these antiherbivore traits.

R: In the literature, the patterns is often reported, but the mechanisms have not been rigorously addressed, as far as we are aware. However, Bell (1982) have suggested an hypothesis in which the patterns are mediated by the effects of climate and especially soil on plant tissues' nutritional quality for herbivores. Our new results are broadly consistent with this hypothesis. Therefore, we have now briefly presented the hypothesis in the Introduction (see third paragraph) and discuss it with detail in the Discussion (see second and fifth paragraphs).

Also, I am curious where dry, nutrient-poor savanna or mesic, nutrient-rich savanna fit into this antiherbivore framework?

R: The SLT is representative of the dry nutrient-rich, whereas the LDW (now ILW) is representative of the mesic nutrient-poor savannas. We have now clarified this in the discussion and now provide a detailed comparison of our antiherbivores with these ecosystems (see fourth, fifth and sixth paragraphs of the discussion).

Line 66 While spinescence is an obvious plant defence trait and leaf size corresponds to your antiherbivores, please justify why wood density would might be associated with plant defence strategies

R: We have now added two sentences to the introduction clarifying the role of wood density as a defence trait:

“Wood density has recently been suggested to be a key antiherbivory resistance trait in savannas, by protecting the stems and branches against breakage and other damage by large herbivores⁷. However, it is less clear whether and how it differs among these antiherbivores.”.

See also our previous paper (Dantas & Pausas 2020).

Line 68 The use of the prehistoric data needs some explanation and critical consideration. On what assumptions is it based, and how reliable is it

R: We have now added more details on this data in the methods section and we also directly discuss potential problems associated with data inaccuracy in the discussion section. However, since most macroecological data also have limitations and draw on several assumptions, we do not feel in position to directly criticize this specific dataset, as this would require some type of validation. Considering this, and the fact that our discussion is already very long, we did not get in much detail. In any case, we are open to make an extra effort if the reviewer believes that this is strictly required.

Line 72: I recommend use of “humans” rather than “men” here and throughout

R: We agree that “humans” is more appropriate and changed it.

Line 81 onwards. It is important that use of body mass is clarified at the start. Mean body mass means mean body mass of megafauna species - it does not (and cannot) account for relative abundance of the species. The authors know this but with our clear explanation it may escape the general reader.

R: You are right. We have now added more detailed information on this index in the last sentence of first paragraph of the Results section.

Line 169 onward - are all paleoclimate models consistent in predicting which areas where savanna in the Pleistocene? Is there any supporting palynological evidence?

R: The study that is cited here is considered a confident source because it uses information from three models simultaneously, rather than only one. This former study also present palynological evidence, some of which are the same as used here to validate the savanna-forest shifts.

Line 480 - More information is needed on the fossil data used to identify the forest areas that were formerly savanna. Can a table of the sites and evidence be provided in the references? I would like to be convinced these were not small patches of savanna in a forest biome (as some are now), or where there is evidence of a shift from savanna-

forest mosaic to continuous closed-canopy forest following megafaunal loss. This is a key part of the argument but is rather rushed in the paper

R: We have now added Table S10 containing details on this fossil data.

The order in which figures in supplementary material are mentioned does not match their sequence of presentation

R: We have now renumbered all the figures. Thank you.

Figure 5 This is potentially a very important figure but I am confused. Five points indicate points that are non forest today, but the salmon points indicate Amazonian savanna patches (and hence also non-forest). If we assume these savanna patches are persistent savanna (perhaps for edaphic reasons) the fossil evidence for grassland to forest transition in Amazonia is limited to periphery of the modern forest biome. The evidence based on the antiherbivore is still there perhaps but this needs some careful discussion. The evidence in the Atlantic forest and caatinga is perhaps stronger.

R: We have now included some more discussion and clarifications on this figure (now Fig. 6)

Figure S1 needs a better caption

R: We have now changed this figure, including more maps and adding a more informative caption. This is now Supplementary Fig. 2.

Table S6, S7 etc Explain the table variables in every caption please

R: We added more detailed explanations to these tables.

References

- Bell, R.H. V. (1982). The effect of soil nutrient availability on community structure in African ecosystems. In: *Ecology of Tropical Savannas* (eds. Huntley, B.J. & Walker, B.H.). Springer-Verlag, New York, pp. 193–216.
- Berzaghi, F., Longo, M., Ciais, P., Blake, S., Bretagnolle, F., Vieira, S., *et al.* (2019). Carbon stocks in central African forests enhanced by elephant disturbance. *Nat. Geosci.*
- Berzaghi, F., Verbeeck, H., Nielsen, M.R., Doughty, C.E., Bretagnolle, F., Marchetti, M., *et al.* (2018). Assessing the role of megafauna in tropical forest ecosystems and biogeochemical cycles – the potential of vegetation models. *Ecography (Cop.)*, 41, 1934–1954.
- Bjorholm, S., Svenning, J., Skov, F. & Balslev, H. (2005). Environmental and spatial controls of palm (Arecaceae) species richness across the Americas, 423–429.

- Bruehlheide, H., Dengler, J., Purschke, O., Lenoir, J., Jiménez-Alfaro, B., Hennekens, S.M., *et al.* (2018). Global trait–environment relationships of plant communities. *Nat. Ecol. Evol.*, 2, 1906–1917.
- Charles-Dominique, T., Davies, T.J., Hempson, G.P., Bezeng, B.S., Daru, B.H., Kabongo, R.M., *et al.* (2016). Spiny plants, mammal browsers, and the origin of African savannas. *Proc. Natl. Acad. Sci. U. S. A.*, 113, E5572–E5579.
- Chave, J., Coomes, D., Jansen, S., Lewis, S.L., Swenson, N.G. & Zanne, A.E. (2009). Towards a worldwide wood economics spectrum. *Ecol. Lett.*, 12, 351–366.
- Coley, P.D. & Barone, J.A. (1996). Herbivory and plant defenses in tropical forests. *Annu. Rev. Ecol. Syst.*, 27, 305–335.
- Dantas, V.L. & Pausas, J.G. (2013). The lanky and the corky: fire-escape strategies in savanna woody species. *J. Ecol.*, 101, 1265–1272.
- Dantas, V.L. & Pausas, J.G. (2020). Megafauna biogeography explains plant functional trait variability in the tropics. *Glob. Ecol. Biogeogr.*, 1–11.
- Diefendorf, A.F., Mueller, K.E., Wing, S.L., Koch, P.L. & Freeman, K.H. (2010). Global patterns in leaf ^{13}C discrimination and implications for studies of past and future climate. *Proc. Natl. Acad. Sci. U. S. A.*, 107, 5738–5743.
- Engemann, K., Sandel, B., Boyle, B., Enquist, B.J., Jorgensen, P.M., Kattge, J., *et al.* (2016). A plant growth form dataset for the New World. *Ecology*, 97, 3243.
- Galetti, M., Mole, M., Jordano, P., Pires, M.M., Paulo, R., Pape, T., *et al.* (2017). Ecological and evolutionary legacy of megafauna extinctions.
- Giacomo, M. Di & Fariña, R.A. (2017). Allometric models in paleoecology : Trophic relationships among Pleistocene mammals, 471, 15–30.
- Gill, J.L. (2014). Ecological impacts of the late Quaternary megaherbivore extinctions. *New Phytol.*, 201, 1163–1169.
- Göldel, B., Araujo, A.C., Kissling, W.D. & Svenning, J.C. (2016). Impacts of large herbivores on spinescence and abundance of palms in the Pantanal, Brazil. *Bot. J. Linn. Soc.*, 182, 465–479.
- Govaerts, R. (2001). How Many Species of Seed Plants Are There?, 50, 1085–1090.
- Hempson, G.P., Archibald, S. & Bond, W.J. (2015). A continent-wide assessment of the form and intensity of large mammal herbivory in Africa. *Science (80-.)*, 350, 1056–1061.
- Kerkhoff, A.J., Moriarty, P.E. & Weiser, M.D. (2014). The latitudinal species richness gradient in New World woody angiosperms is consistent with the tropical conservatism hypothesis, 111, 8125–8130.
- Kier, G., Mutke, J., Dinerstein, E., Ricketts, T.H., Küper, W., Kreft, H., *et al.* (2005). Global patterns of plant diversity and floristic knowledge. *J. Biogeogr.*, 32, 1107–1116.
- Lim, J.Y., Svenning, J.C., Göldel, B., Faurby, S. & Kissling, W.D. (2020). Frugivore-fruit size relationships between palms and mammals reveal past and future defaunation impacts. *Nat. Commun.*, 11, 1–13.

- Liu, H., Gleason, S.M., Hao, G., Hua, L., He, P., Goldstein, G., *et al.* (2019). Hydraulic traits are coordinated with maximum plant height at the global scale. *Sci. Adv.*, 5, eaav1332.
- McNaughton, S.J. (1988). Mineral nutrition and spatial concentrations of African ungulates. *Nature*, 334, 343–345.
- Mendonça, R.C. de, Felfili, J.M., Walter, B., Silva Jr, M., Rezende, A., Filgueiras, T. de S., *et al.* (2008). Flora vascular do bioma Cerrado. In: *Cerrado: Ecologia e Flora* (eds. Sano, S.M., Almeida, S.P. de & Ribeiro, J.F.). Embrapa Informação Tecnológica, Brasília-DF, pp. 421–1279.
- Oliveira-filho, A.T. (2017). *NeoTropTree, Flora arbórea da Região Neotropical: Um banco de dados envolvendo biogeografia, diversidade e conservação*. Univ. Fed. Minas Gerais.
- Onstein, R.E., Vink, D.N., Veen, J., Barratt, C.D., Flantua, S.G.A., Wich, S.A., *et al.* (2020). Palm fruit colours are linked to the broad- scale distribution and diversification of primate colour vision systems.
- Pausas, J.G. & Bond, W.J. (2019). Humboldt and the reinvention of nature. *J. Ecol.*, 107, 1031–1037.
- Pausas, J.G., Keeley, J.E. & Schwilk, D.W. (2017). Flammability as an ecological and evolutionary driver. *J. Ecol.*, 105, 289–297.
- Pimm, S.L. & Joppa, L.N. (2015). How Many Plant Species are There , Where are They , and at What Rate are They Going Extinct? *Ann. Missouri Bot. Gard.*, 100, 170–176.
- Quesada, C.A., Phillips, O.L., Schwarz, M., Czimczik, C.I., Baker, T.R., Patiño, S., *et al.* (2012). Basin-wide variations in Amazon forest structure and function are mediated by both soils and climate. *Biogeosciences*, 9, 2203–2246.
- Ripple, W.J., Newsome, T.M., Wolf, C., Dirzo, R., Everatt, K.T., Galetti, M., *et al.* (2015). Collapse of the world’s largest herbivores. *Sci. Adv.*, 1.
- Wigley, B.J., Fritz, H. & Coetsee, C. (2018). Defence strategies in African savanna trees. *Oecologia*, 187, 797–809.
- Wigley, B.J., Slingsby, J.A., Sandra, D., Bond, W.J., Coetsee, C. & Lyon, C.B. (2016). Leaf traits of African woody savanna species across climate and soil fertility gradients : evidence for conservative versus acquisitive resource-use strategies, 1357–1369.
- Wright, I.J., Cooke, J., Cernusak, L.A., Hutley, L.B., Scalon, M.C., Tozer, W.C., *et al.* (2018). Stem diameter growth rates in a fire-prone savanna correlate with photosynthetic rate and branch-scale biomass allocation, but not specific leaf area. *Austral Ecol.*
- Wright, I.J., Dong, N., Maire, V., Prentice, I.C., Westoby, M., Díaz, S., *et al.* (2017). Global climatic drivers of leaf size, 12, 917–921.
- Zanne, A.E., Lopez-Gonzalez, G., Coomes, D.A., Ilic, J., Jansen, S., Lewis, S.L., *et al.* (2009). Global wood density database. *Dryad*.

Reviewers' Comments:

Reviewer #1:

I enjoyed reading the revision and the response of the authors to my previous comments, they have done a thorough job in addressing those. I am now convinced that the data and sampling are representative for the floras, and analyses and conclusions suitable and supported, and I do not have any further major comments.

Minor:

-Abstract – would it be possible to indicate which traits have been most affected by past megafauna (wood density, leaf spines)? It only says: "We show that megafauna history explains a substantial percentage of defence trait variability." Also, perhaps briefly mention which other non-megafauna variables explain variation in certain traits, such as soil pH (stem spines). I understand that the abstract has to be short, but this would be important information for the reader to obtain here, I think.

Reviewer #2:

Remarks to the Author:

Dear Authors,

Thank you for responding to my questions and critiques.

I am satisfied with your responses and with the added analysis which I think have improved the quality of the paper.

Comments by Reviewer 1

I enjoyed reading the revision and the response of the authors to my previous comments, they have done a thorough job in addressing those. I am now convinced that the data and sampling are representative for the floras, and analyses and conclusions suitable and supported, and I do not have any further major comments.

Minor:

-Abstract – would it be possible to indicate which traits have been most affected by past megafauna (wood density, leaf spines)? It only says: “We show that megafauna history explains a substantial percentage of defence trait variability.” Also, perhaps briefly mention which other non-megafauna variables explain variation in certain traits, such as soil pH (stem spines). I understand that the abstract has to be short, but this would be important information for the reader to obtain here, I think.

R: Following the rationale in the text, we understand and discuss that much of the variability that is explained by soil pH is likely mediated by the effect of soil fertility on herbivore activity. Therefore, we interpret these results as consistent with our initial hypothesis of an herbivore effect, rather than a direct effect of soil on plant traits. Given the limited word count in the abstract, it would be impossible to explain this in detail. Moreover, under this perspective, all of the traits were substantially affected by historical mammal herbivory. Therefore, we implemented the following changes in the abstract: 1) mentioning the effect of soil fertility in the traits (we did not specifically mention pH because, for latex, CEC was more important); 2) mentioning the names of all traits with evidence of being, directly or indirectly, influenced by herbivory (by extant and extinct mammal), and 3) replacing, in the cited sentence, “megafauna history” by “historical mammal herbivory, especially by extinct megafauna” in order to consider that effects mediated by soil fertility may also be mediated by extant mammals. Please, let me know if this is not sufficient.